



# Effects of ship emissions on air quality in the Baltic Sea region simulated with three different chemistry transport models

Matthias Karl[1], Jan Eiof Jonson[2], Andreas Uppstu[3], Armin Aulinger[1], Marja Prank[3,4],
Jukka-Pekka Jalkanen[3], Lasse Johansson[3], Markus Quante[1], and Volker Matthias[1]

[1]Institute of Coastal Research, Helmholtz-Zentrum Geesthacht, 21502 Geesthacht, Germany.
[2]Norwegian Meteorological Institute, Oslo, Norway.
[3]Atmospheric Composition Research, Finnish Meteorological Institute, P.O. Box 503 FI-00101 Helsinki, Finland.
[4]now at: Cornell University, Ithaka, NY, U.S.A..

**Correspondence:** M. Karl (matthias.karl@hzg.de)

**Abstract.** The Baltic Sea is highly frequented shipping area with busy shipping lanes close to densely populated regions. Exhaust emissions from ship traffic into the atmosphere are not only enhancing air pollution, they also affect the Baltic Sea environment through acidification and eutrophication of marine waters and surrounding terrestrial ecosystems. As part of the European BONUS project SHEBA (Sustainable Shipping and Environment of the Baltic Sea Region), the transport, chemical

transformation and fate of atmospheric pollutants in the Baltic Sea region was simulated with three regional chemistry transport models (CTM) systems, CMAQ, EMEP/MSC-W and SILAM with grid resolutions between 4 km and 11 km. The main goal was to quantify the effect that shipping emissions have on the regional air quality in the Baltic Sea region when the same shipping emissions dataset but different CTMs in their typical setups are used. The performance of these models and the shipping contribution to the results of the individual models was evaluated for sulphur dioxide ($SO_2$), nitrogen dioxide ($NO_2$)

and ozone ($O_3$) and particulate matter ($PM_{2.5}$). Model results from the three CTMs were compared to observations from rural and urban background stations of the AirBase monitoring network in the coastal areas of the Baltic Sea region. The performance of the three CTM systems to predict pollutant concentrations is similar. However, observed $PM_{2.5}$ in summer was underestimated strongly by CMAQ and to some extent by EMEP/MSC-W. The spatial average of annual mean $O_3$ in the EMEP/MSC-W simulation is 15–25 % higher compared to the other two simulations, which is mainly the consequence of

using a different set of boundary conditions for the European model domain. There are significant differences in the calculated ship contributions to the levels of air pollutants among the three models. SILAM predicted a much weaker ozone depletion through NO emissions in the proximity of the main shipping routes than the other two models. In the entire Baltic Sea region the average contribution of ships to $PM_{2.5}$ levels is in the range of 4.3–6.5 % for the three CTMs. Differences in ship-related $PM_{2.5}$ between the models are mainly attributed to differences in the schemes for inorganic aerosol formation. Inspection

of the ship-related elemental carbon (EC) revealed that assumptions about the vertical ship emission profile can affect the dispersion and abundance of ship-related pollutants in the near-ground atmosphere. The models are in agreement regarding the ship-related deposition of oxidised nitrogen, reporting a ship contribution in the range of 21–23 ktN y$^{-1}$ as atmospheric input to the Baltic Sea. Results from the present study show the sensitivity of the ship contribution to combined uncertainties of boundary conditions, meteorological data and aerosol formation and deposition schemes. This is an important step towards



a more reliable evaluation of policy options regarding emission regulations for ship traffic and the planned introduction of a nitrogen emission control area (NECA) in the Baltic Sea and the North Sea in 2021.

# 1   Introduction

International shipping is important for the economic exchange in Europe: almost 90 % of the European Union (EU) import and export freight trade is seaborne. Compared to other modes of transport such as trucks and air freight, shipping is far more energy efficient per ton of cargo. The Baltic Sea is one of the most densely trafficked sea regions in the world. Roughly 407,500 ship crossings in the Baltic Sea were recorded in 2012 (HELCOM, 2014), including passenger, cargo, tanker and other ship types. Maritime transport of goods between main EU ports and ports located in the Baltic Sea has a share of 22 % (in 2016) of the total shipping tonnages within European seas (EUROSTAT, 2018).

Ship traffic is associated with exhaust emissions of a wide range of air pollutants, among them nitrogen oxides ($NO_X = NO + NO_2$), black carbon (BC), sulphur dioxide ($SO_2$), non-methane volatile organic compounds (NMVOC) and particulate matter, as well as greenhouse gases (mainly carbon dioxide, $CO_2$). Primary and secondary particulate matter from ship exhaust has been associated with adverse health effects (e.g. Corbett et al., 2007). The emitted amounts and size spectrum of particulate matter depends on the type of fuel and its sulphur content and the ship engine type (e.g. Fridell et al., 2008; Moldanová et al., 2009). For regulatory purposes, particulate matter is divided into the size fractions $PM_{10}$ and $PM_{2.5}$ according to aerodynamic diameter, where $PM_{10}$ covers all particles with diameter less than $10\,\mu m$ and $PM_{2.5}$ those with diameter less than $2.5\,\mu m$. A global model study by Sofiev et al. (2018b) demonstrated the health benefits from reducing the ship-related fine particulate matter by low-sulphur ship fuels in densely populated, major-trading nations.

Emissions from ships are transported in the atmosphere over several hundreds of kilometres (Endresen et al., 2003). The atmospheric transformation of primary emitted gases from shipping is especially relevant for the formation of $O_3$ and the deposition of sulphur and nitrogen compounds distant from the ship lanes (Eyring et al., 2010). Exhaust emissions from ship traffic in the Baltic Sea has the potential to degrade air quality in the coastal areas (Jonson et al., 2015) and to significantly affect the Baltic Sea environment through acidification and eutrophication of marine waters and surrounding terrestrial ecosystems (HELCOM, 2009; Bartnicki et al., 2011; Hunter et al., 2011; Raudsepp et al., 2013; Neumann et al., 2018a, b, c). Acidification is a major challenge in the Baltic Sea region today where the critical load (CL) for acidification is exceeded especially in the southern part (Tsyro et al., 2018). The CL for acid deposition is the value that must not be exceeded in order to maintain the soil's capacity to neutralise the excess acid. Between 2000 and 2016, the percentage of ecosystem areas in Europe where CL (acidification) was exceeded decreased from 16 % to 6 % (Tsyro et al., 2018), partially as a result of stricter sulphur regulations for ships. Despite the considerable improvement concerning critical loads with respect to acidification, there are still regions in the Baltic Sea catchment, for which further reductions in acidification are desirable (Hettelingh et al., 2017). Atmospheric deposition of nitrogen compounds play a role in the eutrophication of the coastal marine environment (e.g., Paerl, 1995) and threaten biodiversity in forests, semi-natural vegetation, and freshwater catchments through excessive nitrogen input (Cofala



et al., 2007). Even though exceedances of CL for eutrophication has decreased over the past decades, critical loads are still exceeded in about 65 % of the European ecosystems (Tsyro et al., 2018).

Air pollution from ships is increasingly controlled worldwide by the International Maritime Organization (IMO) through the Marine Pollution Convention (MARPOL) Annex VI - Regulations for the Prevention of Air Pollution from Ships (IMO, 2008).

The Baltic Sea has been a sulphur emission control area (SECA) since May 2006, with stepwise reductions of the sulphur content in ship fuels; from 2015 onwards the sulphur content of any fuel oil used on-board ships within the Baltic Sea has to be 0.1 % or less (van Aardenne et al., 2013). The MARPOL regulations on emissions of $NO_X$ (TIER) are defined as function of year of installation and vessel speed. The TIER I standard was introduced in 2000 and is up to 10 % stricter than for ships built before 2000. TIER II was implemented in 2011 applying 15 % stricter standards than TIER I. The effect of regulations of

nitrogen emissions have been small until now, as these are only enforced for the new built ships. A nitrogen emission control area (NECA) for the Baltic Sea, North Sea and English Channel will become effective in 2021, in which new built ships have to comply with TIER III, approximately 75 % stricter than TIER II. The consequences of establishing the NECA on the future air quality in the Baltic Sea region are investigated in the companion paper by Karl et al. (2018).

The transport, chemical transformation and fate of atmospheric pollutants in the Baltic Sea region can be simulated with 3-D

chemistry transport model (CTM) systems. Previous air quality model studies related to shipping in the North Sea and Baltic Sea (Matthias et al., 2010; Hongisto, 2014; Jonson et al., 2015, 2018b; Matthias et al., 2016;Antturi et al., 2016; Claremar et al., 2017) used CTM systems to investigate the effect of implementation of MARPOL regulations on sulphur emissions by ships, the effect of establishing the NECA and other ship emission control scenarios. The studies quantified the contributions from shipping to the total air concentrations, deposition of nitrogen and sulphur, as well as air quality and health indicators.

Jonson et al. (2018b) studied the effects of shipping on the global scale, including the effects of shipping in the Baltic Sea and the North Sea, compared to total anthropogenic emissions in a global CTM with $0.5 \times 0.5$ degrees resolution. They found that a significant fraction, ranging from 5 % to more than 10 %, of the $PM_{2.5}$ and the depositions of nitrogen of anthropogenic origin in bordering countries can be attributed to ship emissions in the two sea areas. For the deposition of sulphur, ship emission contributions are markedly lower as a result of the implementation of a stricter SECA (from 2015 onwards). For ground level

$O_3$ contributions were found to vary depending on country and the choice of ozone metric ranging from negative contributions for annual average ozone to more than 10 % of the summertime phytotoxic ozone dose (POD1) for forests in some countries in the Baltic Sea region.

On the regional scale, Jonson et al. (2015) using the EMEP/MSC-W model (Simpson et al., 2012) with a horizontal resolution of $14 \times 14 \, km^2$ assessed the effect of reduced sulphur content (2015 value of 0.1 %) and regulation of NECAs on the air

quality, deposition of nitrogen and related impacts on human health in the Baltic and North seas. Matthias et al. (2016) used the Community Multiscale Air Quality (CMAQ) model v4.7.1 (Byun and Schere, 2006) with a horizontal resolution of $24 \times 24 \, km^2$ to investigate the effects of different future developments of shipping emissions in the North Sea area on air quality in the North Sea region. Antturi et al. (2016) used the SILAM (Sofiev et al., 2015) CTM system with spatial resolution of ca. $8 \times 8 \, km^2$ in a cost-benefit analysis of the sulphur reduction policy in the Baltic Sea, but did not investigate the effects of shipping emissions

on ozone concentration or nitrogen deposition. The study by Claremar et al. (2017) used the EMEP/MSC-W model with a much



coarser resolution ($50 \times 50\,\text{km}^2$) and gridded shipping emissions from the ENTEC inventory (Jonson et al., 2009) for 2011 and from TNO-MACC-III (Gauss et al, 2015) for 2013. They find highest contribution of international shipping in the Baltic Sea and North Sea near large harbour cities and along the main shipping lanes; with contributions to ambient levels of nitrogen dioxide ($NO_2$), $SO_2$, $PM_{10}$, and $PM_{2.5}$ of about 80 %, 80 %, 13 %, and 20 %, respectively (Claremar et al., 2017).

The use of relatively coarse model grids (coarser than 10-km resolution) in some of the previous CTM simulations raises concerns about non-linear chemical effects, particularly for $O_3$, since a high source strength from shipping in the proximity to large land-based emissions (inside the same grid cell of the model) often results in very high levels of $NO_X$ and excessive ozone titration (Jonson et al., 2009). Moreover, shipping releases large amounts of $NO_X$ from a point source within the relatively clean maritime atmosphere. In regional CTMs, these $NO_X$ emissions are diluted into large grid volumes, which can lead to

a systematic overestimation of the ozone production and artificially increases the lifetime of $NO_X$ (Von Glasow et al., 2003; Song et al., 2003; Vinken et al., 2011).

In the Baltic Sea area, movements of ships are gathered in the regional HELCOM (Baltic Marine Environment Protection Commission - Helsinki Commission) Automatic Identification System (AIS) network and database (http://maps.helcom.fi/ website/AISexplorer/). Not all previous air quality studies in the Baltic Sea region used AIS ship position data, some use long

averaging intervals for shipping emissions. For the North Sea region, Matthias et al. (2016) designed emission control scenarios based on a detailed emission inventory built upon AIS ship positions and a detailed ship characteristics database (Aulinger et al., 2016). The studies by Antturi et al. (2016) and Jonson et al. (2015) deployed the Ship Traffic Emission Assessment Model (STEAM; Jalkanen et al., 2009, 2012) which combines the AIS-based information with the detailed technical knowledge of the individual ships.

Despite previous model based work on the effects of shipping on air quality and human health in the Baltic Sea region, there is a need for more localised studies building on a much higher level of details, i.e. concerning shipping activity, for the quantification of regional ship-related air pollution. Knowledge on air quality impacts of shipping with a finer spatial resolution than in previous model studies is required for the identification of best suited sustainable development options for the shipping sector, especially if a varying suite of competing environmental and economic drivers is to be considered in different sub-

regions.

With the goal to support EU policies on environmental and economic aspect of the shipping sector the BONUS project SHEBA (Sustainable Shipping and Environment of the Baltic Sea Region; http://www.sheba-project.eu) was established in 2015. The overarching aim of BONUS SHEBA was an integrated and in-depth analysis of the ecological, economic and social impacts of shipping in the Baltic Sea. The project brought together experts from many different fields (and 11 partner

institutions) in order get a complete picture of the environment impact. Within SHEBA the fate of pollutants into water and air emitted by sea going ships and ships at berth was assessed, both for the present-day situation as well as for several developed scenarios of shipping in the intermediate future. The task was building mainly on a suite of consecutive models following pollutants on the path from emission via dispersion to impacts in target regions.

As part of the SHEBA project, the transport, chemical transformation and fate of atmospheric pollutants in the Baltic Sea

region was simulated with three different regional CTM systems (CMAQ, EMEP/MSC-W and SILAM) to investigate the effect





of ship emissions on the regional air quality in the Baltic Sea region. The chosen CTM systems are well established in Europe and have been extensively tested in several multi-model assessment studies (Solazzo et al., 2012a, 2012b, 2013, 2017; Vautard et al., 2012; Colette et al., 2011, 2012; Langner et al., 2012; Vivanco et al., 2018). Prank et al. (2016) evaluated the skill of air quality models including SILAM, EMEP, and CMAQ to reproduce the particulate matter concentration and composition on

European scale. EMEP/MSC-W (from MET Norway) and SILAM (from FMI) are part of the operational Copernicus Atmosphere Monitoring Service (CAMS; http://www.regional.atmosphere.copernicus.eu/) regional air quality ensemble for Europe. All three models have been used previously in the North Sea and Baltic Sea region for estimating the effect of shipping (Jonson et al., 2015; Antturi et al., 2016; Matthias et al., 2010, 2016). The model setup with CMAQ used in Matthias et al. (2016) has been evaluated for the larger North Sea region (Aulinger et al., 2016).

The EMEP/MSC-W model (at MET Norway) is also included with the same model configuration in the Baltic Sea region Interreg project EnviSuM (Environmental Impact of Low Emission Shipping: Measurements and Modelling Strategies). The main focus of the EnviSuM project is on sulphur emissions from shipping. EnviSuM investigates the effects of the implementation of the stricter SECA from 2015 onwards, combining measurements and modelling. This includes measurements of emissions from ships applying different technologies as for instance low sulphur oil, scrubber technology, and liquefied natural

gas (LNG). The improved emission estimates are used in chemical tracer models calculating the effects on air pollution and depositions and subsequent effects on human health and ecosystems.

This study takes a multi-model approach using three CTM systems to assess the uncertainties connected with the atmospheric transport and transformation of air pollutants. The comparison of air concentration of regulatory pollutants between the models is the primary focus of this study. However, the annual average nitrogen deposition fields were also compared due

to the relevance of atmospheric nitrogen deposition for the eutrophication of the coastal marine environment. In the western Baltic Sea, with its specific coastal features such as boddens and lagoons, the grid resolution of atmospheric CTMs has been found to strongly impact on the modelled nitrogen deposition (Neumann et al., 2018a). Model results from the three CTMs for $NO_X$, $O_3$, $SO_2$ and $PM_{2.5}$ are compared to observations from rural and urban background stations in the coastal areas of the Baltic Sea region. Statistical performance analysis of the comparison of modelled against observation data was carried out for

all CTM systems and the performance of the models was inter-compared based on several statistical indicators. Specifically, we want to evaluate the contribution of shipping emissions to modelled surface concentrations of important air pollutants. The significance of ship contribution to ambient $NO_2$ observations at coastal monitoring stations is evaluated for the different models. The use of three CTM systems, together with comparison to ground-based observations, provides a comprehensive view on the current air quality situation of the Baltic Sea region and how it is affected by emissions from shipping. The combination

of three CTM also provides a more robust estimate of the ship-related contribution to ambient atmospheric concentrations.

## 2 Description of the CTM systems and setup of the model comparison

The setup of the three CTM systems for this study with respect to drivers for meteorology, boundary conditions and emissions was specific for each model system. The models were set up in a way as they are typically used in air quality studies in





European regions. However, the applied CTMs use a much higher spatial and temporal resolution as previous modelling of the air quality in the Baltic Sea region. Shipping activities are considered with a high degree of detail in the simulations; using AIS position data and up-to-date load dependent emission factors for all important air pollutants. The dynamic ship emission inventory STEAM was applied in all CTMs. STEAM takes into account the emission control areas and regulations, emission

abatement equipment on-board the ships as well as fuel sulphur content modelling separately for main and auxiliary engines (Johansson et al., 2017; Jalkanen et al., 2012). All three regional air quality models implement state-of-the-art formulations of atmospheric transport, atmospheric chemistry and aerosol formation, updated compared to the model versions used in the previous studies. Partly the same or similar drivers for anthropogenic emissions were used in the CTMs. Ship exhaust emission from the North Sea are handled in the same manner as the Baltic Sea emissions since they affect the western part of the

Baltic Sea region. By this procedure it is ensured, that all interactions of shipping emissions with pollutants in the regional background and from land-based emission sources are correctly considered. With all models a reference run for the current air quality situation was performed including all emissions ("base") and one run without the emissions from shipping ("noship"). The difference between the run with all emissions and the run without shipping emissions is used to determine the contribution of ships to the ambient pollutant concentration.

## 2.1 Description of the models

### 2.1.1 CMAQ model

The CMAQ model v5.0.1 (Byun and Schere, 2006; Appel et al., 2013; 2017) computes the air concentration and deposition fluxes of atmospheric gases and aerosols as a consequence of emission, transport and chemical transformation. The atmospheric chemistry is treated by the modified Carbon Bond V mechanism cb05tucl with updated toluene chemistry and chlorine radical

chemistry (Yarwood et al., 2005; Whitten et al., 2010; Sarwar et al., 2012). The aerosol scheme AERO5 is used for the formation of secondary inorganic aerosol (SIA). The gas phase/aerosol equilibrium partitioning of sulphuric acid ($H_2SO_4$), nitric acid ($HNO_3$), hydrochloric acid (HCl) and ammonia ($NH_3$) is solved with the ISORROPIA v1.7 mechanism (Fountoukis et al., 2007; Nenes et al., 1999). The formation of secondary organic aerosol (SOA) from isoprene, monoterpenes, sesquiterpenes, benzene, toluene, xylene, and alkanes (Carlton et al., 2010; Pye and Pouliot, 2012) is included.

The dry deposition parameterization is presented in Binkowski and Shankar (1995) and Binkowski and Roselle (2003). Wet deposition of gases and particles is computed by the resolved cloud model of CMAQ which estimates how much certain vertical model layers contributed to the precipitation (Foley et al., 2010). Sea salt emissions were calculated inline by the parameterization of Gong (2003) (as described in Kelly et al., 2010. Sea salt surf zone emissions were deactivated because of considerable overestimations in some coastal regions (Neumann et al., 2016). Biogenic emissions (NMVOC from vegetation

and soil NO) were calculated off-line with the biogenic Emission Inventory System BEIS v3.4 (Schwede et al., 2005; Vukovich and Pierce, 2002). Emissions of wind-blown dust were not considered.

The Meteorology-Chemistry Interface Processor (MCIP; Otte and Pleim, 2010) processes meteorological model output into the input format required for CMAQ. The vertical dimension of the model extends up to 100 hPa in a sigma hybrid pressure





coordinate system with 30 layers. Twenty of these layers are below approximately 2 km; the lowest layer extends to ca. 36 m above ground. A spin-up period of one month (December 2011) was used for the initialization of the model runs, sufficiently long to prevent that initial conditions have an effect on the simulated atmospheric concentrations of the investigated period (year 2012).

### 2.1.2  SILAM model

The SILAM (System for Integrated modeLling of Atmospheric coMposition) model v5.5 (Sofiev et al., 2015; http://silam.fmi.fi/) was used as second CTM in this study. The gas phase chemistry was simulated with the Carbon Bond (CB) mechanism CBM-IV, with reaction rates updated according to the recommendations of the International Union of Pure and Applied Chemistry (IUPAC, http://iupac.pole-ether.fr) and the NASA Jet Propulsion Laboratory (JPL; http://jpldataeval.jpl.nasa.gov) and the terpenes oxidation added from CB05 reaction list (Yarwood et al., 2005). The sulphur chemistry and secondary inorganic aerosol formation is computed with an updated version of the DMAT scheme (Sofiev, 2000) and secondary organic aerosol formation with the Volatility Basis Set (VBS, Donahue et al., 2011), the volatility distribution of anthropogenic organic carbon taken from Shrivastava et al. (2011). Organic aerosol in SILAM was evaluated in a recent study by Prank et al. (2018).

The dry deposition scheme is described in Kouznetsov and Sofiev (2012) and the wet deposition in Kouznetsov and Sofiev (2013). Natural emissions included in the simulations are sea-salt emissions as in Sofiev et al. (2011), biogenic NMVOC emissions as in Poupkou et al. (2010); wild-land fire emissions as in Soares et al. (2015) and wind-blown desert dust.

SILAM includes a meteorological pre-processor for diagnosing the basic features of the boundary layer and the free troposphere from the meteorological fields provided by various meteorological models (Sofiev et al., 2010). In total 10 vertical layers, extending up to 2000 m above the surface, are included. The lowest layer extends to 20 m above the surface. No spin-up period was applied.

### 2.1.3  EMEP/MSC-W model

The third CTM used in this study is the EMEP/MSC-W chemical transport model, version rv4.8. This model, available as open source (https://github.com/metno/emep-ctm), has been described in detail in Simpson et al. (2012), with various updates, see Simpson et al. (2016) and references within. Chemistry scheme of the gas-phase in the model is EmChem09, having 72 chemical compounds including 10 "surrogate" VOCs, out of which isoprene represents BVOCs, and 137 reactions. This scheme is an update of previous chemical schemes (e.g. Simpson, 1992; Andersson-Sköld and Simpson, 1999). The EMEP scheme involves relatively more details on peroxy radical ($RO_2$) chemistry than e.g. Carbon Bond (CB) schemes. SOA is calculated using a VBS scheme, which tracks the semi volatile products of VOC oxidation, and dynamically partitions these between the gas and aerosol phases (e.g. Robinson et al., 2007). A number of schemes were tested in Bergström et al. (2012), but here the standard "NPAS" scheme as described in Simpson et al. (2012) is used.

For Europe the model is regularly evaluated against measurements in the EMEP annual reports, see www.emep.int. In addition the EMEP model has been included in model inter-comparisons and model validations in a number of peer reviewed publications (Jonson et al., 2006, 2010, 2018a; Simpson et al., 2006a, 2006b; Colette et al., 2011, 2012; Angelbratt et al., 2011;



Dore et al., 2015; Stjern et al., 2016). Biogenic emissions (NMVOC, soil NO), emissions of dimethyl sulphide (DMS) from oceans, sea-salt, dust, road dust, emissions from aviation on cruising altitude, lightning, volcanic emissions and emissions from forest fires are included as separate databases or calculated within the model (EMEP, 2015).

EMEP is driven by the meteorological data of the European Centre for Medium-Range Weather Forecasts (ECMWF) based
on the Cy40r1 version. An important addition to the forecast ensemble in cycle Cy40r1 has been the introduction of ocean-atmosphere coupling from day 0, instead of from day 10 as in the previous cycles. Vertically, the meteorological fields from ECMWF are interpolated onto 20 EMEP sigma levels, between the surface and 100 hPa. Initial and boundary concentrations are based on long-term observations and some model data. No spin-up period was applied.

## 2.2   Setup of the CTM systems

The CTMs were offline coupled with different meteorological models (COSMO-CLM, ECMWF-IFS, and WRF). CMAQ and SILAM were operated with high horizontal resolution (4 km) on the inner nest representing the Baltic Sea region starting from simulations of a coarser European domain. EMEP MSC-W was operated on 0.1 degree resolution for the whole of Europe. Ship emissions from the STEAM model (Jalkanen et al., 2009; Jalkanen et al., 2012 Johansson et al., 2013), which use ship position data of the HELCOM AIS network, were gridded to the respective model's grid resolution. Land-based emissions
were from SMOKE-EU (Bieser et al., 2011a) or ECLIPSE (Amann et al., 2012; Amann et al., 2013) databases; annual totals were comparable.

### 2.2.1   Model domains and nesting

The spatial extent for the intercomparison study covers the Baltic Sea region, spanning from latitude 53.50°N (south) to 66.00°N (north) and longitude 9.00°E (west) to 31.00°E (east). Parts of the Kattegat and a small part of the Norwegian Sea
is covered by the extent, but not considered in the comparison. The extent of the geographic domain is displayed in Fig. 1. Nested simulations were done with CMAQ and SILAM models, using the output of the finer resolved inner nest whereas the simulation with the EMEP/MSC-W model covered the European domain. The SILAM model was operated on rotated grids centred on the respective domain. The horizontal grid resolution of the output was 4 km for CMAQ, 0.04 degrees (~4 km) for SILAM and 0.1 degrees (~11 km) for EMEP/MSC-W. Note that the grid distance in x-direction becomes smaller with
increasing latitude (for instance, 0.1° in longitude corresponds to 6.2 km at 56°N). Different drivers were used for meteorological simulations coupled offline to the CTM simulations. Anthropogenic emissions from the continent and the shipping emissions in the North and Baltic seas were identical (CMAQ and SILAM) or similar in spatial distribution and magnitude (EMEP/MSC-W). The EMEP/MSC-W model used monthly averaged gridded ship emissions, while the other two models used hourly emissions. Daily or hourly emissions reflect ship traffic pattern changes due to meteorological conditions or due to
sea ice. Using a coarser time resolution for shipping thus mainly neglects the influence of weather and ice on ship operations (Jonson et al., 2015). Table 1 gives an overview of the model setups of the three CTM systems for use in the intercomparison study.

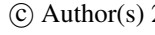



Nested simulations with CMAQ were performed with a coarse outer domain for entire Europe with grid cell size of $64 \times 64\,km^2$, an intermediate domain with $16 \times 16\,km^2$ for Northern Europe and an inner domain with a horizontal resolution of $4 \times 4\,km^2$ for the entire Baltic Sea. Model results for the intercomparison were taken from the inner domain for the coastal regions and from the intermediate domain for parts of Sweden, Finland and the Baltic states. For details on the high-
resolution output from CMAQ and an evaluation of the model setup with a limited number of regional background stations, it is referred to Karl et al. (2018).

For the SILAM model, the grid cell size was roughly $70 \times 70\,km^2$ for the outer domain, roughly $18 \times 18\,km^2$ for the central domain, and roughly $4 \times 4\,km^2$ for the inner domain. The simulation time steps were 20 min, 10 min, and 4 min, respectively. Model results for the intercomparison were mostly from the inner domain, with parts of Finland and eastern Europe from the
central domain. SILAM took part in AQMEII 1 and 3 intercomparisons showing comparable performance with other European state-of-the-art air quality models (Solazzo et al., 2012a, 2012b, 2013, 2017; Vivanco et al., 2018; Marécal et al., 2015). The EMEP/MSC-W model was run with a $0.1 \times 0.1$ degrees resolution for whole Europe. A comprehensive description, including model evaluations, of the model results with the $0.1 \times 0.1$ degrees application of the EMEP model for 2013 can be found in Tsyro et al. (2015).

### 2.2.2 Meteorology

The SILAM model is run with meteorological input from a simulation with the Weather Research and Forecast (WRF) model v3.7.1 using original resolutions of 4.0 km, 16.0 km, and 64.0 km, for inner, central and outer domains, respectively. In general, linear interpolation was applied for the simulation, but conservation of mass was used where applicable. High-resolution meteorological fields for CMAQ were obtained from the COSMO-CLM (Rockel et al., 2008) model v5.0. More
details on the meteorological forcing data and the evaluation of precipitation can be found in Karl et al. (2018). The meteorological fields were converted to the extension, resolution and projection of the CMAQ nested grids, using an in-house modified version of MCIP. EMEP/MSC-W was driven by meteorological data from the Integrated Forecasting System (IFS) of the ECMWF, version IFS38r2, with t1279 resolution (about 0.16 degrees resolution) interpolated to $0.1 \times 0.1$ degrees.

The use of different meteorological datasets introduces additional variability which is on one hand wanted to achieve a wider
range of possible results for estimating the effect of shipping on air quality but on the other hand complicates the interpretation of differences between the models.

### 2.2.3 Boundary conditions

The initial conditions (ICONs) for the simulation and the lateral boundary conditions (BCONs) for the outer European domain are taken from FMI APTA global reanalysis (Sofiev et al., 2018a)). The global boundary conditions results have been inter-
polated in time and space to provide hourly boundary conditions for the respective outer domains of the CMAQ and SILAM simulations. The setup for initial and boundary concentrations for EMEP/MSC-W is described in Simpson et al. (2012). ICONs and BCONs are based on long-term observations. For ozone, 3-D fields for the whole domain are specified from climatological ozone sonde datasets, modified monthly against clean-air surface observation. For most other chemical compounds they are



defined by simple functions based on measurements and/or model calculations, prescribing concentrations in terms of latitude and time-of-year, or time-of-day.

### 2.2.4 Anthropogenic land-based emissions

Anthropogenic land-based emissions in hourly resolution obtained from the SMOKE-EU (Bieser et al., 2011a) emission in-
ventory were provided for CMAQ and SILAM. These emissions are based on officially reported EMEP emissions which are then distributed in time and space using appropriate surrogates like population density maps, street maps or land use maps. Point sources from the European point source emission register are considered. Vertical distribution of point source emissions is based on real-world stack information and calculated within SMOKE-EU (Bieser et al., 2011b). Dynamic emissions from agricultural activity and animal husbandry depending on meteorological variability are considered (Backes et al.,
2016). EMEP/MSC-W model uses anthropogenic emissions from the ECLIPSE emission inventory (http://www.iiasa.ac.at/web/home/research/researchPrograms/air/ECLIPSEv5.html). These emissions differ slightly from the reported national total EMEP emissions for 2012, see Wankmüller and Mareckova (2014). For the countries bordering the Baltic Sea (excluding Russia) the national total sulphur emissions from ECLIPSE are about 6 % higher and the $NO_2$ emissions about 10 % lower than the corresponding EMEP emissions.

### 2.2.5 Shipping emissions

Shipping emissions for the Baltic Sea and North Sea areas were obtained from STEAM (Jalkanen et al., 2009, 2012; Johansson et al., 2013, 2017). The shipping emissions are used together with the land-based emissions described in Sect. 2.2.4 in all three CTM systems. STEAM uses ship position data of individual ships collected from the AIS base stations network. The AIS data is combined with technical information of the specific vessels and engine types to derive emissions for each individual ship.
Shipping emissions are calculated dynamically to consider the emission control areas and regulations, emission abatement equipment on-board the ships as well as fuel sulphur content modelling, separately for main and auxiliary engines (Johansson et al., 2017; Jalkanen et al., 2012).

The STEAM inventory for the Baltic Sea shipping emissions used in the SHEBA project, consist of hourly updated $2 \times 2\,\mathrm{km}^2$ gridded data for $NO_X$, $SO_X$, carbon monoxide (CO), and particulate matter, which is further divided into elemental carbon,
organic carbon, sulphate ($SO_4^{2-}$) and mineral ash. For the North Sea and other European seas the STEAM data for 2011 were used. Ship emission were used with hourly time resolution in CMAQ and SILAM whereas they were used with monthly resolution in EMEP/MSC-W.

The use of monthly aggregated ship emissions in EMEP/MSC-W is justified by the fact that the same set of ship emissions from FMI is applied for different meteorological years in the routine application of EMEP modelling and that ship emissions
from other seas were only available for 2011. Previous tests with daily and monthly aggregated ship emissions showed that the differences in results are very small. The use of North Sea ship emissions from 2011 on hourly basis in CMAQ and SILAM causes some inconsistency because meteorological data of 2012 is used in the CTM simulations. Because we are



mainly interested in the seasonal variability of pollutant concentrations based on daily averages, the outcome of this study will be less affected by the inconsistency between the timing of ship emissions and the meteorological conditions.

STEAM emission data was provided for two vertical layers (below 36 m, from 36–1000 m). The vertical distribution of STEAM emissions was applied differently in the CTMs (see Table 2). In the CMAQ model, emissions below 36 m were
attributed to the lowest vertical model layer, while emissions above 36 m were attributed to the second lowest layer. In the SILAM simulation, the emissions were assumed to be evenly distributed within the emission layers between ground and 1000 m, and were attributed to the model layers accordingly. In EMEP/MSC-W all ship emissions were attributed to the lowest vertical model layer, which typically has a height of 92 m.

## 2.3   Statistical analysis

### 2.3.1   Evaluation method

Model results for surface concentrations of $NO_2$, $O_3$, $SO_2$, and $PM_{2.5}$ from the three CTMs are evaluated against available measurements of the air quality monitoring network from the AirBase version 8 database (Simoens, 2014). AirBase is the air quality information system maintained by the European Environmental Agency (EEA) through the European topic centre on Air Pollution and Climate Change Mitigation.

Table S1 gives a list of all rural and regional background monitoring stations. Concentrations of $NO_2$ are monitored at 17 stations, $O_3$ at 35, $SO_2$ at 11, and $PM_{2.5}$ at 8 rural/regional background stations. Table S2 gives all urban and suburban background monitoring stations included in the statistical evaluation of the models. Concentrations of $NO_2$, $O_3$, $SO_2$, and $PM_{2.5}$ are monitored at 52, 46, 37, and 10 stations of the urban background, respectively. Fig. 2a shows locations of stations with $NO_2$ and with $O_3$ measurements. Fig. 2b shows locations of the stations with $SO_2$ and with $PM_{2.5}$ measurements.

The model output of surface concentration fields of each CTM is used with its original horizontal resolution to calculate daily mean concentrations. The modelled concentrations are extracted from the respective model grid cell where the selected monitoring stations are located. The evaluation was done for the entire year 2012 based on daily means. The model output for $PM_{2.5}$ was taken from the modelled $PM_{2.5}$ containing aerosol water at 50 % relative humidity.

The performance of each model is quantified in terms of mean values ($\mu_{\mathrm{Mod}}$ and $\mu_{\mathrm{Obs}}$), normalized mean bias (NMB),
Spearman's correlation coefficient (R), root mean square error of the modelled values (RMSE) and fraction of model values within factor 2 of the observations (FAC2). The model performance analysis is discussed separately for rural background stations and urban background stations. In order to better highlight model differences in terms of urban areas and station types (i.e. rural, sub-urban, urban background sites), groups of stations (rural versus urban) are generated in which statistical performance indicators are averaged. In the rural group, rural background and regional background stations are included, while
in the urban group, urban background and sub-urban background stations are included. Monitoring stations classified as traffic stations and industrial stations were not included in the comparison, since the regional CTM systems applied here do not handle the local scale dispersion near emission sources.





In the context of this evaluation of predicted air pollutant concentrations, we consider a correlation coefficient of more than 0.5 to indicate a correlation between modelled and observed time series, while values of 0.7 and above are considered as a good correlation. Hanna and Chang (2012) define certain acceptance criteria for model performance based on their experience in conducting a large number of model evaluation exercises. For rural stations FAC2 values > 0.5 and for urban stations FAC2 values > 0.3 indicate acceptable performance. We adopt these bounds in the present study to characterize the predictive strength of the models with respect to the pollutant concentrations.

Further, we compare the performance between models with the help of a graphical comparison in form of boxplots. Boxplots of the correlation coefficient, NMB and RMSE including either all rural or all urban monitoring stations were prepared. The boxplots show the median as line dividing the box in two parts, the upper and lower quartiles as end of the box, the minimum and maximum values of the data and outliers.

### 2.3.2 Significance of the ship contribution

The method described in Aulinger et al. (2016) was used to assess the significance of ship influence on ambient $NO_2$ at the monitoring stations. The ship influence at a station was positively confirmed in the tests if: (1) the concentrations increased and (2) the temporal correlation improved, when shipping emissions are included in the CTM simulation.

By means of a paired t-test it was first tested whether the modelled $NO_2$ concentrations at the monitoring stations with available $NO_2$ observations (Table S1) significantly increased if shipping emissions were considered. This test estimated whether the mean concentration difference between the "noship" run and the "base" run ("noship" - "base") is significantly equal to or greater than zero, indicated by the probability $p_{bias}$. If the value of $p_{bias}$ is larger than the level of significance of 0.05, than this hypothesis was confirmed. Otherwise, it was decided that the model run without shipping emissions led to lower concentrations, confirming the ship influence.

The significance of the improvement in the correlation between simulations and observations was tested by calculating the Fisher z transformation of the two correlation coefficients for the two model runs ("noship" and "base") and testing the hypothesis "greater than". Correlation coefficients were calculated with Spearman's method (Myers and Sirois, 2006) for consistency with the statistical evaluation. The probability $p_{corr}$ for the hypothesis that the correlation between the base run and observations is greater than the correlation between the "noship" run and observations was calculated. We accepted this hypothesis if the probability was higher than 0.9. Therefore, in the following, a station $i$ with $p_{corr,i} > 0.9$ for a specific CTM simulation is termed ship-influenced.

## 3 Results

### 3.1 Comparison to observations

A statistical performance analysis for each of the three CTMs was undertaken using the available observation data form AirBase for 2012 based on daily mean concentrations. The results of the statistical performance analysis are summarized in Table S3





for daily mean NO$_2$, in Table S4 for daily mean O$_3$, in Table S5 for daily mean SO$_2$ and in Table 3 for daily mean PM$_{2.5}$. In the following, the performance of the models to simulate air pollutant concentrations is compared and discussed separately for the group of rural stations and for the group of urban stations in order to highlight differences in the predictive capability of the models for rural versus urban sites. Moreover, the significance of the ship contribution to the monitored daily mean NO$_2$ has

been tested for the three CTMs. The results of the significance test are summarized in Table S6.

### 3.1.1  Statistical evaluation for NO$_2$ at rural and urban sites

Emissions of NO$_X$ from traffic and various combustion sources are mainly in the form of nitrogen oxide (NO) which oxidises to NO$_2$ within a few minutes through the titration reaction with O$_3$. NO$_2$ plays a major role in atmospheric reaction cycles that produce ground-level ozone. NO$_2$ also contributes to the formation of secondary particulate matter. The atmospheric lifetime

of NO$_2$ is relatively short; a few hours in summer and up to one day in winter (Schaub et al., 2007), hence differences between rural and urban sites are expected due to the higher emission density in urban or industrial areas.

Fig. S1 compares the times series of daily mean NO$_2$ concentrations from the three CTMs at two rural background stations and at two urban background stations. The seasonal variation at the two rural sites DESH008 (Bornhöved, Germany) and DK0054A (Keldsnor/9054, Denmark) with peak concentrations in winter are captured by the models (Fig. S1a,b). CMAQ

shows frequent peaks at the DK0054A during summer. This feature is also seen for other coastal rural sites and could indicate that, due to the high spatial resolution of the model, ship exhaust plumes are resolved but not adequately dispersed. Since the models do not specifically treat the plume dispersion of individual ships, the spreading of the plume might not be sufficiently large or the plume rise of ship exhaust is not properly considered with the applied vertical profile of ship emissions. The other two models also tend to give higher NO$_2$ than observed at DK0054A in summer. The correlation between predicted and

observed daily mean NO$_2$ for this site is low for all models (R < 0.3). The seasonal variation at the two urban sites DK0053A (Aalborg/8158, Denmark) and LV0RKE2 (Riga Kengarags-2, Latvia) are reproduced by the models, although CMAQ and SILAM underestimate daily mean NO$_2$ in winter at LV0RKE2 (Fig. S1c,d).

The rural station average of observed annual mean NO$_2$ is 6.9 µg m$^{-3}$ (range: 1.1–15.8 µg m$^{-3}$). The models reproduced the annual means at rural sites very well; modelled annual mean NO$_2$ is 6.5, 8.1 and 6.0 µg m$^{-3}$ for CMAQ, SILAM and

EMEP/MSC-W, respectively. The overall correlation of NO$_2$ for rural stations is good for all models, with R of 0.72, 0.64 and 0.65 for CMAQ, SILAM and EMEP/MSC-W, respectively, as shown in the boxplots in Fig. 3a. At most rural stations the NMB is between -0.5 and 0.5 for all models. An exception is SE066A (Norr Malma, Sweden), where NO$_2$ is largely overestimated by the models (outlier in the boxplot). This is also the only rural station, for which FAC2 of all models is below 0.5. For SE066A, the influence of traffic emissions from the nearby motorway might be too high in the models, as the spatial resolution is still

not fine enough to resolve the dispersion near local sources. The average RMSE at rural sites is 4.7, 5.9 and 3.9 µg m$^{-3}$ for CMAQ, SILAM and EMEP/MSC-W, respectively.

The urban station average of observed annual mean NO$_2$ is 16.7 µg m$^{-3}$ (range: 6.7–34.1 µg m$^{-3}$), more than twice the concentration average at rural sites. The three CTMs underestimate the annual means at urban sites (range of model averages: 9.9–10.9 µg m$^{-3}$). The overall correlation of NO$_2$ for urban sites is lower than at the rural sites, with R between 0.5 and 0.6 for





the models, as shown in the boxplots in Fig. 3b. At most urban stations, models underestimate the observed $NO_2$ by ca. 40 %
(NMB ranges: -0.9–0.1, -0.8–0.2 and -0.8–0.4 for CMAQ, SILAM and EMEP/MSC-W, respectively) (Table S3). For some
urban stations the FAC2 is below 0.3, but for the majority of the stations all models fulfil the acceptance criteria for urban sites.
The average RMSE at urban sites is 9.9, 10.1 and 9.6 µg m$^{-3}$ for CMAQ, SILAM and EMEP/MSC-W, respectively.

The general underestimation of $NO_2$ at urban sites has been evident in other multi-model air quality studies in Europe (e.g.
Giordano et al., 2015). The finer horizontal resolution of CMAQ and SILAM (4 km) compared to EMEP/MSC-W (11 km)
does not result in a significant improvement of the urban bias and urban temporal correlation. This result was expected based
on the study by Schaap et al. (2015), who found no further improvement of the urban signal, i.e. the concentration difference
between high emission areas and their surroundings, when increasing the resolution from 14 km to 7 km in the same model.
Moreover, increasing the spatial resolution in the model does not help to improve significantly the performance in time because
the temporal variability of pollutants is largely controlled by the meteorological conditions and pollution levels upwind (Schaap
et al., 2015).

### 3.1.2   Statistical evaluation for $O_3$ at rural and urban sites

Tropospheric ozone is largely controlled by the atmospheric transport from regions outside the study area, by stratosphere-
troposphere exchange and by the photochemical production through the oxidation of VOCs and carbon monoxide (CO) in the
presence of $NO_X$ and sunlight. Near the surface, the depletion via reaction with NO (titration reaction) in environments with
high $NO_X$ levels and dry deposition are important sinks of ozone.

The seasonal variation at the two rural sites DESH008 and DK0054A (same stations as for $NO_2$) is captured by the models,
but high $O_3$ concentrations in July and August at DK0054A are underestimated by CMAQ and SILAM (Fig. S2a,b). The
underestimation of observed ozone in summer by the CMAQ simulation is also found for other coastal stations (Karl et al.,
2018) and indicates too low photochemical reactivity. The seasonal variation at the two urban sites DK0053A and LV0RKE2
is captured by the models (Fig. S2c,d). The good performance of the models for $O_3$ at the urban sites is partly attributed to the
high spatial resolution, as $NO_X$ emissions are injected into a smaller grid box volume and consequently less diluted initially.
Thus the models are capable of representing the titration effect on the day-to-day variation of ozone.

The rural station average of observed annual mean $O_3$ is 56 µg m$^{-3}$ (range: 44–71 µg m$^{-3}$). The models show a good correla-
tion with observed $O_3$ at the rural stations is, with R of 0.74, 0.76 and 0.77 for CMAQ, SILAM and EMEP/MSC-W (Fig. 4a).
Modelled daily mean $O_3$ is in agreement with observed daily mean $O_3$ within a narrow NMB range for rural sites. CMAQ
and SILAM have no bias (average NMB = 0.0), whereas EMEP/MSC-W slightly overestimates the measurements (average
NMB = 0.17). The ozone bias might be linked to boundary conditions (Giordano et al., 2015): EMEP/MSC-W uses ozone
boundary conditions from long-term observations, whereas CMAQ and SILAM models use boundary conditions from the
FMI APTA global reanalysis.

The higher density of $NO_X$ emissions in urban areas is expected to lead to a larger titration effect of NO on ozone, which
results in lower average $O_3$ at the urban sites compared to rural sites. The urban station average of observed annual mean $O_3$
is 48 µg m$^{-3}$ (range: 38–59 µg m$^{-3}$). CMAQ and SILAM predict similar annual mean concentrations as observed for both rural





and urban sites, whereas EMEP/MSC-W predicts higher annual mean ozone. As for the rural sites, the models show a good correlation with observed $O_3$ at the urban stations, with R between 0.73 and 0.77 (Fig. 4b). The models slightly overestimate the $O_3$ measurements at urban sites, with CMAQ having the smallest bias (average NMB = 0.08).

FAC2 of the models is larger than 0.75 for all rural sites and larger than 0.57 for all urban sites. The average RMSE values for the rural sites and the urban sites, respectively, are similar for the three models (Fig. 4), indicating comparable model performance for the CTMs with respect to daily mean $O_3$ concentrations.

### 3.1.3 Statistical evaluation for $SO_2$ at rural and urban sites

Another major air pollutant is $SO_2$. It is primarily emitted from anthropogenic emission sources such as coal power plants, residential heating, waste incineration and shipping activities. $SO_2$ acts as a precursor to sulphates, which are one of the main components of particulate matter in the atmosphere. The atmospheric lifetime of $SO_2$ is on the order of a few days (Lee et al., 2011). $SO_2$ can still be considered to be relatively short-lived and thus less influenced by transport from regions outside the study area. Most emission sources of $SO_2$ are located in urban areas. In the case of power plants, the emissions of $SO_2$ are however injected at elevated height and therefore do not directly impact the surface concentrations in the urban area. On the other hand, residential heating emissions of $SO_2$ strongly influence surface concentrations in urban areas during winter. In coastal harbour cities, $SO_2$ from shipping is a relevant contributor (Viana et al., 2014).

The seasonal variation at the two rural sites SE0001A (Rådhuset, Sweden) and FI00208 (Luuki, Finland; rural) is rather well reproduced by the models (Fig. S3a,b). CMAQ and SILAM overestimate the observed $SO_2$ concentrations in winter. EMEP/MSC-W underestimates observed $SO_2$ concentrations at FI00208 and also at other rural stations. The seasonal variation at the two urban sites DESH023 (Lübeck-St. Jürgen, Germany) and LT00044 (Kaunas-Noreikiskes, Lithuania) is captured by the models (Fig. S3c,d). CMAQ and SILAM show too high $SO_2$ concentrations in winter and EMEP/MSC-W underestimates observed $SO_2$ during summer.

The rural station average of observed annual mean $SO_2$ is 1.3 µg m$^{-3}$ (range: 0.7–2.5 µg m$^{-3}$). The annual means of the models at rural sites is in the range of 0.8 to 2.2 µg m$^{-3}$. With only 10 stations, the rural station group for $SO_2$ is rather small, limiting the conclusions that can be drawn from the statistical analysis. The model data and the observations for $SO_2$ at the rural stations are correlated, with R of 0.67, 0.65 and 0.55 for CMAQ, SILAM and EMEP/MSC-W (Fig. 5a). At rural and urban sites, the modelled daily mean $SO_2$ from CMAQ and SILAM has a positive bias, whereas modelled daily mean $SO_2$ from EMEP/MSC-W has a slight negative bias. FAC2 of the models is larger than 0.5 for most rural sites. For CMAQ, only one station (EE0016A, Saarejärve, Estonia) does not fulfil the acceptance criteria, due to frequent overestimation of observed $SO_2$. The average RMSE at rural sites is 1.5, 2.4 and 1.2 µg m$^{-3}$ for CMAQ, SILAM and EMEP/MSC-W, respectively.

At urban stations, the correlation between model data and observed $SO_2$ shows a mixed performance of the models, with good correlation at some stations and poor correlation at others (Fig. 5b). The urban station average of observed annual mean $SO_2$ is 3.9 µg m$^{-3}$ (range: 1.0–8.0 µg m$^{-3}$), three times higher than at the rural stations. CMAQ and SILAM predict similar annual mean concentrations, whereas EMEP/MSC-W predicts 36 % lower concentrations for the urban sites. For CMAQ and SILAM, FAC2 is larger than 0.3 for all urban sites. For EMEP/MSC-W some stations have a FAC2 below 0.3, due





to underestimation of observed $SO_2$. The average RMSE at urban sites is 4.2, 4.3 and 3.3 µg m$^{-3}$ for CMAQ, SILAM and EMEP/MSC-W, respectively.

### 3.1.4 Statistical evaluation for PM$_{2.5}$ at rural and urban sites

Ambient PM$_{2.5}$ is a wide-spread pollutant, which is directly emitted by biomass and fossil fuel combustion in domestic and
industrial activities, and it is also formed from gaseous precursors such as $NO_X$, $SO_2$, $NH_3$ and NMVOC in the atmosphere. The atmospheric lifetime of PM$_{2.5}$ is on the order of days or weeks and thus PM$_{2.5}$ can be subject to long-range transport.

The seasonal variation at the two rural sites SE0001A and FI00208 (same stations as for $SO_2$) is well reproduced by the models (Fig. S4a,b). CMAQ and EMEP/MSC-W underestimate the observed PM$_{2.5}$ concentrations at SE0001A in summer. The seasonal variation at the two urban sites DESH023 and LT00044 is nicely reproduced by the models (Fig. S4c,d). CMAQ
underestimates the observed PM$_{2.5}$ concentrations at both urban stations in summer. This is also found for other stations of the regional background and can partly be attributed to the underestimation of secondary organic aerosols and to the missing emissions of wind-blown soil dust particles in the CMAQ simulation (Karl et al., 2018).

The rural station average of observed annual mean PM$_{2.5}$ is 7.0 µg m$^{-3}$ (range: 3.3–12.6 µg m$^{-3}$). The modelled annual mean PM$_{2.5}$ is 5.5, 6.7 and 3.8 µg m$^{-3}$ for CMAQ, SILAM and EMEP/MSC-W, respectively. For PM$_{2.5}$ smaller differences
between rural and urban stations are expected than for $NO_2$ and $SO_2$ because PM$_{2.5}$ has a large secondary component, which is generally more homogeneously distributed over rural and urban areas. The urban station average of observed annual mean PM$_{2.5}$ is 10.5 µg m$^{-3}$ (range: 4.8–13.2 µg m$^{-3}$) within a similar range as the rural sites. SILAM predicts a similar annual mean for urban stations, whereas CMAQ and EMEP/MSC-W give lower annual mean values.

For rural stations the temporal correlation of PM$_{2.5}$ is slightly better for EMEP/MSC-W than for CMAQ and SILAM
(Fig. 6a). For urban stations the temporal correlation of PM$_{2.5}$ is good, with average R between 0.68 and 0.71 (Fig. 6b). At rural and urban sites, the modelled daily mean PM$_{2.5}$ from CMAQ and EMEP/MSC-W has a slightly negative bias, whereas modelled daily mean PM$_{2.5}$ from SILAM has no bias (Table 3).

The average RMSE at the rural sites is between 4.9 and 5.4 µg m$^{-3}$. It should be noted that RMSE station values for SILAM are within a smaller range (between lower and upper quartile) than the other two models. The performance for urban stations
in terms of RMSE is similar for the models, with average RMSE between 6.4 and 7.2 µg m$^{-3}$.

### 3.1.5 Significance of ship contribution to NO$_2$ at monitoring stations

A statistical method (Sect. 2.3.2) was applied to decide whether the modelled concentration as well as the correlation with observed $NO_2$ concentration at a specific station increases significantly when ship emissions are included in the CTM simulation. If both the concentration and the correlation increased, the station is termed ship-influenced. A significant concentration in-
crease was found at all 69 stations for CMAQ and EMEP/MSC-W but only at 78 % of the stations for SILAM. However, the significance of the concentration increase only shows that the modelled concentrations at a station are sensitive to ship emissions. The correlation increases significantly (on 0.9 or 0.95 level) at 7, 10, and 11 stations for CMAQ, SILAM and EMEP/MSC-W, respectively (Table S6).





The ship-influenced stations identified by CMAQ included two stations at the shoreline (EE0011R, Vilsandi, Estonia, Fig. S5a; FI00349, Utö, Finland, Fig. S5b), three stations in small harbour cities (DESH023, Lübeck-St. Jürgen; EE0022A, Narva, Estonia; SE0022A, Södermalm, Sweden) and two stations further inland (FI00431, Palokka 2, Finland; PL0077A, KpZielBoryTuch, Poland). Some of these stations (EE0011R, FI00349, FI00431 and PL0077A) were also identified by the
other models. The ship influence at station Rostock Warnemünde (DEMV021, Germany), located close to a harbour, was significant in both EMEP/MSC-W and SILAM but not in CMAQ (Fig. S5c). This could indicate that differences in the meteorological data, in particular wind flow fields, are responsible for the different ship influence. Although, the timing and location of ship exhaust plumes - based on AIS data - should be accurate during the port stays, the emission fluxes at berth are more challenging to estimate, because this involves estimation of electrical power usage during the port stays. Station DK0054A
(Keldsnor/9054) (Fig. S5d), although close to the shoreline, did not fulfil the correlation criteria, either because the station is not situated downwind of the ship plumes or the models did not properly resolve the ship plumes due to the sub-grid variability of the plume dispersion (see Sect. 3.1.1).

## 3.2    Comparison of the spatial distribution of air quality indicators

### 3.2.1    Spatial distribution of annual mean $NO_2$

A strong south-north gradient for annual mean $NO_2$ concentrations is found for the Baltic Sea region in the three "base" simulations, with 4–5 times higher $NO_2$ concentrations in the south-western part than in the northern part of the region (Fig. 7a). High modelled $NO_2$ concentrations are predicted in Denmark, northern Germany, and Poland as well as over the Danish Straits and in the urbanized areas of the region. Modelled annual mean $NO_2$ concentrations in proximity of the main shipping routes several times exceed the concentrations in the regional background. EMEP/MSC-W shows the strongest concentration
gradients between urban and rural areas and between ship lanes and surrounding sea. The simulations with the other two models result in a wider spread of the $NO_X$ emissions from the ship routes and the urban centres, indicating stronger horizontal transport by advection and diffusion in CMAQ and SILAM. This finding is counter-intuitive as the $NO_X$ emissions are initially less diluted than in the EMEP/MSC-W simulation because of the smaller volume of the grid boxes and should therefore result in higher $NO_X$ concentrations near the emission sources.

Atmospheric transport by diffusion processes are sub-grid mixing processes, which are not resolved by the given resolution of the applied models. For large grid cells, e.g. $50 \times 50 \, km^2$, the numerical diffusion will usually be much larger than the physical diffusion in the horizontal direction. However, at finer resolution scales, the physical diffusion will gradually become more important than numerical diffusion and becomes greater than numerical diffusion for $5 \times 5 \, km^2$ cell size or below (Karl et al., 2014).

The wider spread of elevated $NO_2$ concentrations is also indicative for a longer atmospheric lifetime of $NO_2$ in CMAQ and SILAM compared to the simulation with EMEP/MSC-W. $NO_2$ is removed relatively quickly in the lower troposphere through the reaction with hydroxyl (OH) radicals to form $HNO_3$. The rate coefficient for this reaction, $k(NO_2+OH)$, is similar in the three models ($(1.1–1.2) \times 10^{-11} \, cm^3 \, s^{-1}$ at 298 K). Thus, differences in the $NO_2$ lifetime are mainly due to different abundances



of OH radicals in the simulations. High $NO_2$ concentrations in Belarus and Russia in the SILAM simulation are an artefact from merging with the output of the coarser central model grid (Sect. 2.2.1).

### 3.2.2 Spatial distribution of annual mean $O_3$

Modelled annual mean $O_3$ concentrations over the Baltic Sea are 15–25 % higher than over land, but are reduced along the ship lanes due to the titration effect caused by the ship-emitted $NO_X$ (Fig. 7b). Lowest ozone concentrations are seen for St. Petersburg ($< 32\,\mu g\,m^{-3}$) in the three simulations, although we note that the city is outside of the high-resolution grid in the case of SILAM. The spatial average of annual mean $O_3$ is clearly higher for the EMEP/MSC-W simulation, by 15–25 %, compared to the other two simulations (Table 4). The most probable reason for the difference is the application of different sets of boundary conditions for the European model domains, as discussed in Sect. 3.1.2. Model simulations for Europe have shown a high sensitivity of ozone changes to the dry deposition to vegetation (Andersson and Engardt, 2010). Thus differences in the deposition schemes may partly explain the different $O_3$ levels over the continent, e.g. when comparing ozone over Sweden and Finland between CMAQ and SILAM.

### 3.2.3 Spatial distribution of annual mean $SO_2$

Clear differences in the spatial distribution of the annual mean $SO_2$ concentrations are found between CMAQ and SILAM on one hand and EMEP/MSC-W on the other hand (Fig. 7c). The simulation with CMAQ and SILAM show a southeast-northwest gradient with elevated $SO_2$ over large parts of the southern Baltic Sea region, Poland, Belarus, Russia and the Baltic States with annual mean concentrations in the range of $1.3$–$3.0\,\mu g\,m^{-3}$. Residential heating emissions and power plant emissions for district heating in the urban centres and rural areas strongly contribute to the high $SO_2$ concentrations in this sub-region. In the EMEP/MSC-W simulation, elevated $SO_2$ concentrations are present along the main shipping routes, in urban areas and in Poland, whereas the levels of $SO_2$ outside of these areas are much lower.

The concentration gradients between urban and rural areas and between ship lanes and surrounding sea is up to $2.5\,\mu g\,m^{-3}$ for EMEP/MSC-W while it is only up to $0.7\,\mu g\,m^{-3}$ for CMAQ and SILAM. Factors contributing to the different gradients are differences in the representation of horizontal transport (see Sect. 3.2.1), differences in the meteorological conditions, and differences in the atmospheric lifetime of $SO_2$. The atmospheric lifetime of $SO_2$ is determined by its reaction with the OH radical and by its removal via dry deposition. In EMEP/MSC-W, the canopy uptake of $SO_2$ is strongly controlled by $NH_3$ levels, and the implemented deposition parameterization accounts for co-deposition effects on the dry deposition of $SO_2$ (Simpson et al., 2012). Co-deposition effects are not considered in the other two models.

### 3.2.4 Spatial distribution of annual mean $PM_{2.5}$

Modelled annual mean $PM_{2.5}$ is higher in the southern part, both over land and sea, than in the northern part of the Baltic Sea region (Fig. 7d). On annual average, $PM_{2.5}$concentrations are not elevated along the ship routes. The seasonal differences between summer and winter will be discussed below (Sect. 3.4) and will help to understand differences between the models.





High PM$_{2.5}$ levels (8 µg m$^{-3}$ and higher) are simulated in the urban areas of major cities like Copenhagen, Oslo, Helsinki, Riga, Tallinn and St. Petersburg. The high PM$_{2.5}$ levels over the continent in the southern part of the Baltic Sea region presumably result from a combination of land-based primary emissions, long-range transported particles and the formation of secondary particulate matter.

### 3.3 Comparison of the ship contribution in the three CTMs

The influence of shipping emissions on the air quality was evaluated for the annual mean concentrations of the three CTMs. The results for the impact of shipping emissions were calculated as differences between the "base" and the "noship" simulations. Results for the absolute ship-related concentrations of O$_3$, NO$_2$, SO$_2$ and PM$_{2.5}$ are shown in Figure 8, the resulting relative ship contribution to annual mean concentrations is shown in Fig. S6 and the spatial average of the relative ship contribution is given in Table 5.

#### 3.3.1 Ship contribution to annual mean NO$_2$

The ship-related annual mean NO$_2$ concentrations from the three CTMs are in the range of 3–5 µg m$^{-3}$ along the main ship routes. The NO$_2$ ship contribution decreases to Baltic Sea background values (about 1 µg m$^{-3}$) within a few hundred kilometres distance from the centre of the shipping routes. Ships emit NO$_X$ mainly in the form of NO, which is however quickly converted to NO$_2$, thus atmospheric NO$_X$ is mainly in the form of NO$_2$. The relative contribution of ship emissions to annual mean NO$_2$ is more than 40 % over the Baltic Sea (Fig. S6) and 22–28 % for the entire Baltic Sea region (Table 5). NO$_X$ emissions from shipping affect the harbour cities of the region and coastal areas in southern Sweden. SILAM predicts the weakest relative ship influence in the Baltic Proper and the Gulf of Finland. Such local differences might be due to the different meteorological drivers or differences in the titration efficiency for ozone.

#### 3.3.2 Ship contribution to annual mean O$_3$

In the proximity of the main shipping routes, negative concentration differences for the modelled annual mean O$_3$ between the "base" and the "noship" simulation are obtained as a result of the titration effect by the NO$_X$ emissions from shipping. The highest ozone reduction due to shipping is found in the western part of the Baltic Sea. In the CMAQ simulation the depletion of ozone is stronger than in the other two models; with O$_3$ reduction by 6–12 µg m$^{-3}$ in the Kattegat and in the Danish Straits. The hourly variation of ship emissions is represented in the simulations with CMAQ and SILAM, whereas monthly averaged ship emissions are used in the EMEP/MSC-W simulation. Emission peaks of NO$_X$ from ships that are present in the hourly data can result in occasional stronger ozone titration leading to overall higher reduction of ozone, than it is the case for monthly averaged ship emissions. The SILAM simulation shows only a small influence of shipping on annual mean O$_3$ along the ship lanes, i.e. the titration effect is much weaker than in the other two models. SILAM shows ozone increases by ca. 2 µg m$^{-3}$ in a distance of several hundred kilometres from the ship lanes in Baltic Proper, which is not evident for the other models.





This might indicate a $NO_X$-limited regime in some areas of the Baltic Sea, where ozone increases with increasing $NO_X$ from shipping.

### 3.3.3 Ship contribution to annual mean $SO_2$

Ship emissions of $SO_2$ have a high contribution to annual mean $SO_2$ concentrations over the Baltic Sea. The ship contribution to
$SO_2$ is 0.5–0.7 $\mu g\,m^{-3}$ in a wide corridor around the main shipping routes of the Baltic Sea. While the absolute ship contribution of the three CTMs is similar, the relative ship contribution in the EMEP/MSC-W simulation is higher in most areas of the Baltic Sea and in Sweden, because the background atmospheric $SO_2$ levels in this simulation are lower than in CMAQ and SILAM.

### 3.3.4 Ship contribution to annual mean $PM_{2.5}$

The ship contribution to annual mean $PM_{2.5}$ shows a gradient from southwest to north with highest concentrations over Den-
mark, the west coast of Sweden, the Belt Sea/Kattegat and over the sea south of Sweden with maximum values up to 0.9 $\mu g\,m^{-3}$. The relative contribution in these ship-impacted areas is up to 10 %. In the entire Baltic Sea region the average contribution is in the range of 4.3–6.5 % for the three CTMs. The absolute ship contribution in SILAM is smaller than for the other two models, in particular in the southwest part of the Baltic Sea region (Fig. 8d). A possible explanation for this is the lower dry deposition of fine aerosols in SILAM, in particular over seawater surfaces (Kouznetsov and Sofiev, 2012). The ship-related $PM_{2.5}$ affects
the coastal areas in the Baltic Sea region, as its influence extends further inland than it is the case for ship-related $NO_2$ or $SO_2$. This can be attributed to the formation of secondary particulate matter in the ship exhaust plume during its transport away from the main shipping routes.

### 3.4 Comparison of $PM_{2.5}$ in summer and autumn

CMAQ and EMEP/MSC-W simulations predict higher seasonal mean concentrations of $PM_{2.5}$ in autumn (average of Septem-
ber, October and November, SON) than in summer (average of June, July and August, JJA), whereas the SILAM simulation predicts higher $PM_{2.5}$ in summer (Fig. 9a,c; Table S9). The temporal correlation between model data and observations of daily mean $PM_{2.5}$ for the average of the AirBase stations is slightly better in autumn than in summer (Table S7 and S8). Observed $PM_{2.5}$ in summer is underestimated strongly by CMAQ (at all stations) and to some extent by EMEP/MSC-W. In autumn all models are in better agreement with observed $PM_{2.5}$.
In summer, modelled mean $PM_{2.5}$ in the region is much higher in the SILAM simulation (5–8 $\mu g\,m^{-3}$ in most parts; 7.4 $\mu g\,m^{-3}$ on domain average) than for the other two models (< 4 $\mu g\,m^{-3}$, except for the urban areas). The higher summertime $PM_{2.5}$ in SILAM is most likely due to more efficient SOA production and different primary emission from wildfires and/or mineral dust. A previous comparison of the models to $PM_{2.5}$ observations from the EMEP station network in Europe reported similar seasonal mean concentrations of the SIA components, i.e. nitrate ($NO_3^-$), ammonium ($NH_4^+$) and $SO_4^{2-}$, for the three
CTMs in summer (Prank et al., 2016).





The calculated ship contribution from all models is higher in summer than in autumn (Table S9). The simulations reflect the greater importance of shipping activities during summer and their influence on PM$_{2.5}$ levels over the entire Baltic Sea and the coastal areas (Fig. 9b). In particular Denmark and the Swedish west coast is highly impacted in summer, with a ship contribution of 0.5–0.9 µg m$^{-3}$ to ambient PM$_{2.5}$ levels.

In autumn, all CTMs predict high levels of PM$_{2.5}$ in the southern part of the Baltic Sea region, exceeding 6 µg m$^{-3}$ . High PM$_{2.5}$ in autumn is typically attributed to stagnant meteorological conditions and higher emissions of primary particulate matter from residential heating and energy production. Modelled PM$_{2.5}$ in Sweden and Finland is higher in SILAM than in the other two models. SILAM overestimates observed PM$_{2.5}$ at the stations in Sweden, Lithuania and Finland in summer (NMB: 0.54 on average) and autumn (NMB: 0.31 on average). In an earlier model comparison, all three models were shown to overestimate

NO$_3^-$ and NH$_4^+$ in autumn; while SILAM also overestimated SO$_4^{2-}$ in autumn (Prank et al., 2016).

The ship contribution in autumn in the southwest part of the region is higher in EMEP/MSC-W compared to the other two models (Fig. 9d), obviously a result from larger secondary formation of particulate matter, as mainly the coastal regions are impacted. The formation of SIA in autumn is favoured by lower temperature and higher humidity compared to summer. Since land-based anthropogenic emissions of NH$_3$ and NO$_2$ as well as the ship emissions are the same in CMAQ and SILAM, the

lower autumn ship contribution in SILAM is mainly attributed to differences in the schemes for inorganic aerosol formation. The investigation of differences between the SIA formation schemes is, however, beyond the scope of this study.

### 3.5 Comparison of elemental carbon related to ship emissions

Primary carbonaceous particles emitted from ships are the product of incomplete fuel combustion and consist of a mixture of elemental carbon and non-polar organic carbon. In the STEAM ship emission inventory these are separated into emissions of

elemental carbon (EC) and organic carbon (OC). The terms EC and BC are used interchangeably in the models, however both can only be regarded as proxies for the concentration of soot particles (Vignati et al., 2010). Here we are mainly interested in the atmospheric fate of EC from ship emissions, as simulated by the models. EC particles are associated with adverse human health effects (Dockery et al., 1993) and contribute to regional haze and poor visibility (e.g. Odman et al., 2007). The atmospheric lifetime of EC is relatively long; around 6 days in the continental outflow (Park et al., 2005) and 4–8 days on global scale

(Vignati et al., 2010), with a large uncertainty due to soot ageing processes and wet deposition (Textor et al., 2006). Therefore the removal by deposition within the study region is expected to be rather limited.

The spatial averages of the EC concentrations ("base" simulation) and the ship-contributed EC concentrations are given in Table 6. The seasonality of ship-related EC predicted by the three CTMs is shown in Fig. 10. The levels of ship-related EC are higher in spring and summer than in autumn and winter due to more intense shipping activities. Therefore the ship contribution

peaks in the seasons when ambient EC concentrations are lowest (Table 6; Fig. S7). The highest levels of ship-related EC, in the range of 0.03–0.04 µg m$^{-3}$, occur along the main shipping routes and in the main ports of the region. On regional average, ships contribute 5–7 % to the annual mean EC (Table 5). SILAM predicts a stronger seasonal variability of the ship-related EC than the other models. In particular, modelled EC ship contribution is spring and winter are lower than for CMAQ and EMEP/MSC-W.





Shipping emissions of EC are identical in the three CTMs on a monthly basis. Differences between the models are therefore explained by differences in the meteorological conditions, the treatment of atmospheric transport in the models and the vertical ship emission profiles. In CMAQ and EMEP/MSC-W all ship emissions are injected below 100 m height. The lower surface concentrations of ship-related EC in EMEP/MSC-W can thus be attributed to different atmospheric stability, i.e. more frequent

neutral conditions, which dilutes the concentration of the emitted pollutant. The fact that EC is more confined to the shipping routes and shows a limited spatial spreading compared to the other models might also indicate less efficient horizontal diffusion in EMEP/MSC-W.

In SILAM, a vertical ship emission profile that specifically distributes the STEAM emissions from 0 m up to 1000 m above the ground, to mimic the plume rise of ship exhaust (Table 2). This way, typically 38 % of the ship emissions are injected

in a height above 500 m in the model. In stable conditions, the boundary layer (BL) height over the Baltic Sea is often at or below 500 m (Svensson et al., 2016; Gryning and Batchvarova, 2002), which would effectively prevent pollutant mass emitted in higher vertical layers to reach the surface layer. Climatological simulations over the Baltic Sea show that there is a strong seasonality in the atmospheric stability over the sea with more than 50 % stable conditions in spring whereas during the other seasons unstable conditions dominate together with occasionally neutral conditions (Svensson et al., 2016). Stable

conditions over the Baltic Sea in the SILAM simulation will reduce the quantity of ship emissions that can affect the surface layer concentrations. Thus in SILAM a larger proportion of the ship emissions becomes subject to long-range transport and is exported out of the Baltic Sea region. In winter, SILAM and EMEP/MSC-W ship-related EC is very low and the impacted area has a smaller extent than in the other months (white areas in Fig. 10), indicating faster removal of EC particles than in the CMAQ simulation. Different treatment of the hygroscopicity and ageing processes of EC particles, affecting their wet

scavenging, could have contributed to the differences among the models.

### 3.6 Comparison of oxidised nitrogen deposition

Annual atmospheric deposition of total nitrogen to the Baltic Sea basin has declined by about one third between 1995 (305 kt y$^{-1}$) and 2015 (222 kt y$^{-1}$) (Bartnicki et al., 2017). During the two decades, the deposition of oxidised nitrogen ($N_{oxi}$) decreased by 35 % whereas the deposition of reduced nitrogen ($N_{red}$), i.e. $NH_3$ and $NH_4^+$, decreased by only 12 % (Bartnicki

et al., 2017).

Results of the spatial distribution of oxidised nitrogen deposition from the three models were compared (Fig. S8). The annual deposition of $N_{oxi}$ shows a strong gradient from southwest to northeast in the simulation by all models. Deposition of $N_{oxi}$ to the Baltic Proper is larger in SILAM and EMEP/MSC-W (280–350 mg(N) m$^{-2}$) than in CMAQ (230–290 mg(N) m$^{-2}$). The companion study by Karl et al. (2018) reports a wide-spread underestimation of the wet deposition of nitrate and ammonium

by CMAQ in 2012 compared to observations of the regional background monitoring stations of the EMEP network. The underestimation of nitrogen deposition in CMAQ is likely caused by a combination of too slow oxidative conversion of $NO_X$ to $HNO_3$, low particulate ammonium from the regional background and underestimation of precipitation in the southern Baltic Sea (Karl et al., 2018).





The ship-related annual deposition of oxidised nitrogen is however similar among the models and on average 40–60 mg(N) m$^{-2}$ over the Baltic Sea. The relative contribution of shipping emissions to the deposition of N$_{oxi}$ to the Baltic Sea is 24 % for CMAQ (Karl et al., 2018) and 14–18 % for the other two models. The contribution from ships to the oxidised nitrogen deposition in the Baltic Sea basin for year 2012 in the present study is therefore higher than the previous reported range from 12 to 14 % (Hongisto, 2014) for the period 2008 to 2011. The ship-related contribution to the nitrogen deposition to the Baltic Sea is in the range of 21–23 ktN y$^{-1}$ for the three CTMs of this study (Fig. 11), thus 40–70 % higher than for single years of the Hongisto (2014) study. The higher ship contribution in the present study might partially be due to the inter-annual variation of nitrogen deposition caused by changing meteorological conditions, which is typically in the range of -13 % to 17 % (Hongisto, 2014). Most of northern Europe experienced above-average precipitation during 2012; in Finland the annual precipitation in 2012 was 739 mm, which is 173 mm above the 1961–1990 average (WMO, 2013). The Hongisto (2014) study applied the STEAM ship emission inventory and fine resolution grids (0.068 degrees), however the specific treatment of SIA formation or the abundance of atmospheric oxidants and NH$_3$ represent possible explanations for the discrepancy.

The total nitrogen deposition to the Baltic Sea when all emissions are considered is ~230 ktN y$^{-1}$ for EMEP/MSC-W and SILAM, comparable to the estimate of 223.6 ktN y$^{-1}$ (2012; normalised by inter-annual changes in meteorological conditions) used in the HELCOM evaluation of the Baltic Sea marine environmental status (Bartnicki et al., 2017) which uses the EMEP model with the coarser 50 km-resolution. CMAQ computes lower deposition totals: N$_{oxi}$ is 25 % lower and N$_{red}$ is 40 % lower compared to the other two models (Fig. 11). According to the SILAM simulation, wet deposition has a share of 60 % and 66 %, respectively, to the total deposition of N$_{oxi}$ and N$_{red}$ to the Baltic Sea. The wet deposition share was similar in the EMEP simulation but lower in the CMAQ simulation (N$_{oxi}$: 54 %; N$_{red}$: 63 %). For SILAM and EMEP/MSC-W, the wet deposition share to total nitrogen deposition is within the range of 63–70 % given for the years 1995–2006 (Bartnicki et al., 2011).

## 4  Summary and conclusions

The effect of ship emissions on the regional air quality and nitrogen deposition in the Baltic Sea region was investigated with three regional CTM systems (CMAQ, EMEP/MSC-W and SILAM) that simulate the transport, chemical transformation and fate of atmospheric pollutants. The models were applied with their typical setup for air quality studies in European regions. The use of three different CTM systems has the advantage of providing the necessary information for assessing the uncertainties connected with the atmospheric transport and transformation of pollutants. The same ship emission dataset from the STEAM model based on ship movements from AIS records, detailed ship characteristics and up-to-date load dependent emission factors were used in all CTMs. The models were set up with a finer grid resolution (4-km to 11-km grid length) than it was the case for previous air quality studies in the Baltic Sea region, potentially enabling a better treatment of the dispersion and photochemistry in exhaust plumes from shipping along the major ship lanes of the Baltic Sea.

The comparison of air concentrations of regulatory air pollutants among the models is the primary focus of this study. Results from the three CTMs were compared to observations from rural and urban background stations of the AirBase monitoring network in the coastal areas of the Baltic Sea region. The performance of the models to predict pollutant concentrations was





found to be similar. The finer resolution of CMAQ and SILAM (4 km) compared to EMEP/MSC-W (11 km) did not lead to a significant improvement of the urban bias and urban temporal correlation for daily mean $NO_2$ concentrations. The benefit from using high-resolution grids depends on the availability of accurate urban emission data with high spatial resolution (Schaap et al., 2015) and realistic temporal profiles (Kukkonen et al., 2012). While the STEAM inventory provides this data for

shipping, the compilation of urban emission inventories is more challenging because they are based on specific information for each sector, such as housing units for domestic heating or number of vehicles (Guevara et al, 2016). Observed $PM_{2.5}$ in summer is underestimated strongly by CMAQ at all stations and to some extent by EMEP/MSC-W. In autumn all models are in better agreement with observed $PM_{2.5}$. The low summer $PM_{2.5}$ in CMAQ has been attributed to the underestimation of secondary organic aerosols and to the missing emissions of wind-blown soil dust particles (Karl et al., 2018). Particulate matter emissions

from wind-blown dust and forest fires are included in EMEP/MSC-W and SILAM.

The ship influence on $NO_2$ concentrations at coastal monitoring stations was investigated with a statistical method. A station is regarded as ship-influenced if the modelled concentration as well as the correlation with observed $NO_2$ increased significantly (on 0.9 level or higher) when ship emissions are included in the CTM simulation. Ship-influenced stations identified by the models are mainly located close to the shoreline or close to a port; but a few inland stations up to 200 km distant from the shore

were also found to be influenced from shipping. However, modelled peaks of high daily mean $NO_2$ at coastal rural sites during summer that are not present in the measurements indicate that the models often did not properly resolve the ship plumes due to the sub-grid variability of the plume dispersion of individual ships.

The spatial average of annual mean $O_3$ concentrations in the EMEP/MSC-W simulation is 15–25 % higher compared to the other two simulations. EMEP/MSC-W overestimates the measurements of daily mean $O_3$ concentrations at rural stations by

17 % on average. The higher ozone concentrations in the EMEP model are mainly the consequence of using a different set of boundary conditions for the European model domain. The concentration gradients of $NO_2$ and $SO_2$ between urban and rural areas and between ship lanes and the surrounding sea are larger in EMEP/MSC-W than in the other models. Factors contributing to the different gradients are differences in the representation of horizontal transport, differences in the meteorological driving data, and differences in the atmospheric lifetime of $NO_2$ and $SO_2$ in the models.

There are significant differences in the calculated ship contributions to the levels of air pollutants among the three models. In the proximity of the main shipping routes, ozone is depleted as a result of the titration effect by NO emissions from shipping. SILAM predicted a much weaker titration effect than the other two models. In contrast with the other models, SILAM shows ozone increases due to shipping several hundred kilometres away from the shipping lanes in the Baltic Proper. The ship-related $PM_{2.5}$ affects the coastal areas in the Baltic Sea region, as its influence extends further inland than it is the case for ship-related

$NO_2$ and $SO_2$. In the entire Baltic Sea region the average contribution of ships to levels of $PM_{2.5}$ is in the range of 4.3–6.5 % for the three CTMs. Differences in ship-related $PM_{2.5}$ among the models are mainly attributed to differences in the schemes for inorganic aerosol formation.

Since shipping emissions of elemental carbon are identical in the three CTMs on a monthly basis, differences for ship-related EC can be explained by differences in the meteorological conditions, the treatment of atmospheric transport in the models and




the vertical ship emission profiles. In SILAM a large proportion of the ship emissions is injected in altitudes above 500 m and becomes subject to long-range transport out of the Baltic Sea region.

The ship-related annual deposition of oxidised nitrogen is similar among the models. The annual oxidised nitrogen deposition from ships to the Baltic Sea is on average 40–60 mg(N) m$^{-2}$ and annual totals are in the range of 21–23 ktN y$^{-1}$, and thus higher

than previous estimates. The higher ship contribution in the present study might partly be a result of the high precipitation during 2012.

Results obtained from the use of three CTMs give a more robust estimate of the ship contribution to atmospheric concentrations and deposition than a single model. By using several models the sensitivity of the ship contribution to uncertainties of boundary conditions, meteorological data as well as aerosol formation and deposition schemes is taken into account. This is

an important step towards a more reliable evaluation of policy options regarding emission regulations for ship traffic and the introduction of a nitrogen emissions control area (NECA) in the Baltic Sea and the North Sea in 2021. In order to improve the calculation of the ship emission influence on local and regional air quality further, better constraints for the vertical emission profile from ships are needed. The plume rise from moving ships should be studied under atmospheric stability conditions relevant in the European seas with the aim to derive generally applicable parameterisations for the vertical profile.

*Data availability.* Data from the simulations with CMAQ, SILAM and EMEP/MSC-W on air pollutant concentrations and nitrogen deposition are available upon request.

*Author contributions.*

**Matthias Karl:** overall structure; CMAQ simulations; evaluation of the air concentration data from all models; framework for statistical processing; visualisation and plotting; main writing tasks

**Jan Eiof Jonson:** Simulations with the EMEP/MSC-W model; model data provision; text contributions

**Andreas Uppstu:** SILAM simulations; model data provision; model description

**Armin Aulinger:** statistical method for ship contribution; R-Scripts for visualisation and data processing

**Marje Prank:** SILAM simulations; discussions of model results

**Jukka-Pekka Jalkanen:** shipping emissions with STEAM; text contributions

**Lasse Johansson:** shipping emissions with STEAM

**Markus Quante:** development of research questions; discussions of the manuscript structure; text contributions to the introduction, conclusions and the abstract

**Volker Matthias:** development of research questions; coordination of the intercomparison

*Competing interests.* The authors declare that they have no conflict of interest





*Acknowledgements.* This work resulted from the BONUS SHEBA project and was supported by BONUS (Art 185), which is jointly funded by the EU, the Academy of Finland and Forschungszentrum Jülich Beteiligungsgesellschaft mbH. The contributions from the EMEP model were funded by the European Union (European Regional Development Fund) project EnviSuM and partially by EMEP under UNECE. The Baltic Sea Regional Development Fund is acknowledged for financial support.

5    We are grateful to the HELCOM member states for allowing the use of HELCOM AIS data in this research. We thank Jana Moldanová (IVL) and Malin Gustafsson (IVL) for providing WRF data that was used in the SILAM simulation. Johannes Bieser (HZG) is thanked for providing SMOKE-EU emission datasets. Computer time for EMEP model runs was supported by the Research Council of Norway through the NOTUR project EMEP (NN2890K) for CPU, and NorStore project European Monitoring and Evaluation Programme (NS9005K) for storage of data. The air quality model CMAQ is developed and maintained by the U.S. Environmental Protection Agency (US EPA). COSMO-

10   CLM is the community model of the German climate research (www.clm-community.eu). The simulations with COSMO-CLM and CMAQ were performed at the German Climate Computing Centre (DKRZ) within the project "Regional Atmospheric Modelling" (Project Id 0302). The European Environmental Agency (EEA) is thanked for the AirBase air quality database system.



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





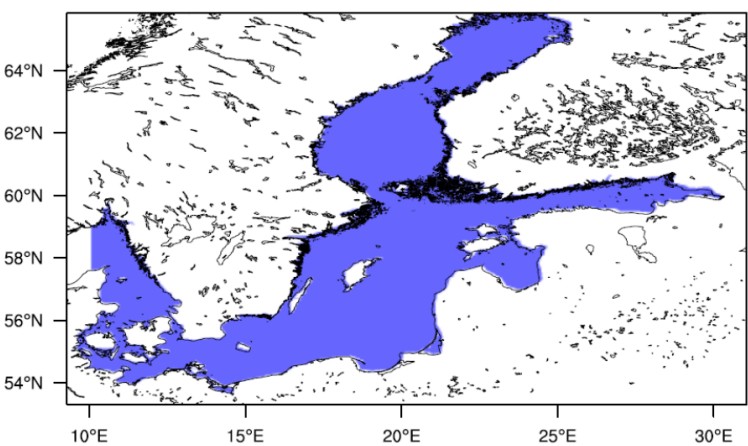

**Figure 1.** Geographic map of the study domain for the CTM comparison, spanning from latitude 53.40°N (south) to 65.80°N (north) and longitude 9.00°E (west) to 31.10°E (east). The extent of the Baltic Sea as used in this study is shown in blue.

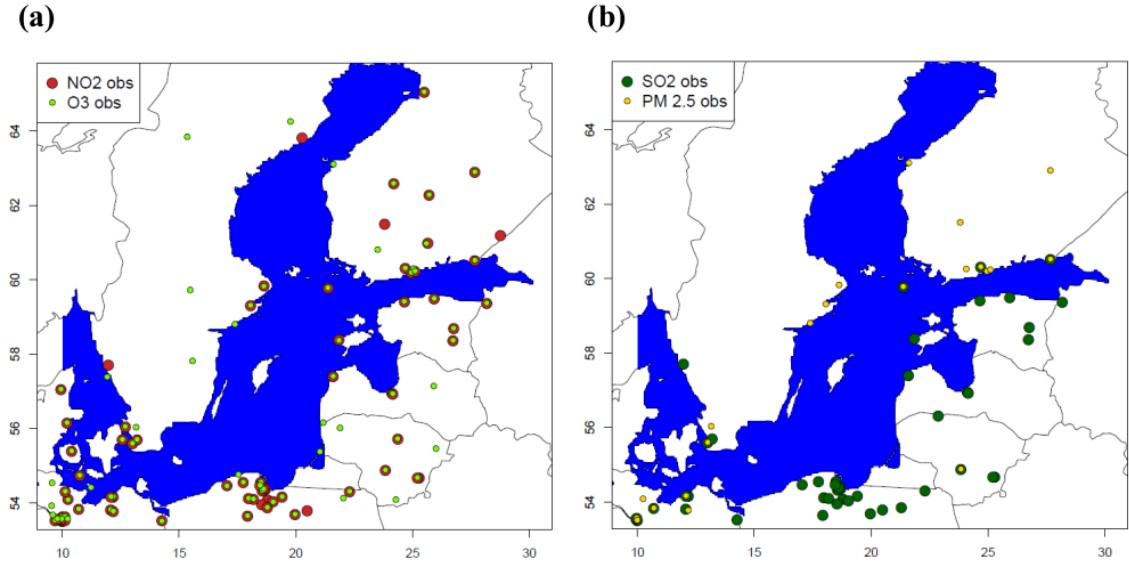

**Figure 2.** Map of the Baltic Sea region with the location of background monitoring stations used in the statistical performance analysis with observations of: (a) $NO_2$ (filled red circles) and $O_3$ (filled green circles); and (b) $SO_2$ (filled dark green circles) and $PM_{2.5}$ (filled yellow circles). Same domain extent as in Figure 1.





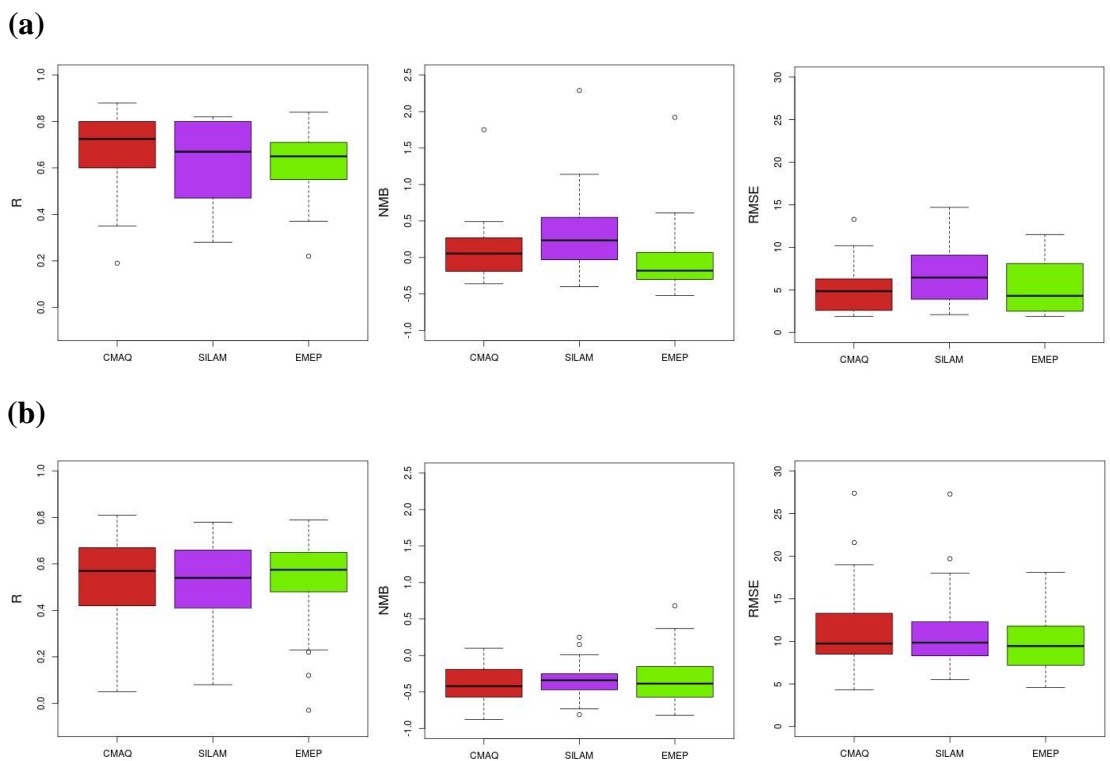

**Figure 3.** Comparison of statistical indicators for NO$_2$ daily means (in the order R, NMB, and RMSE) between three CTMs at: (a) rural background stations and (b) urban background stations. Outlier shown as small circles.





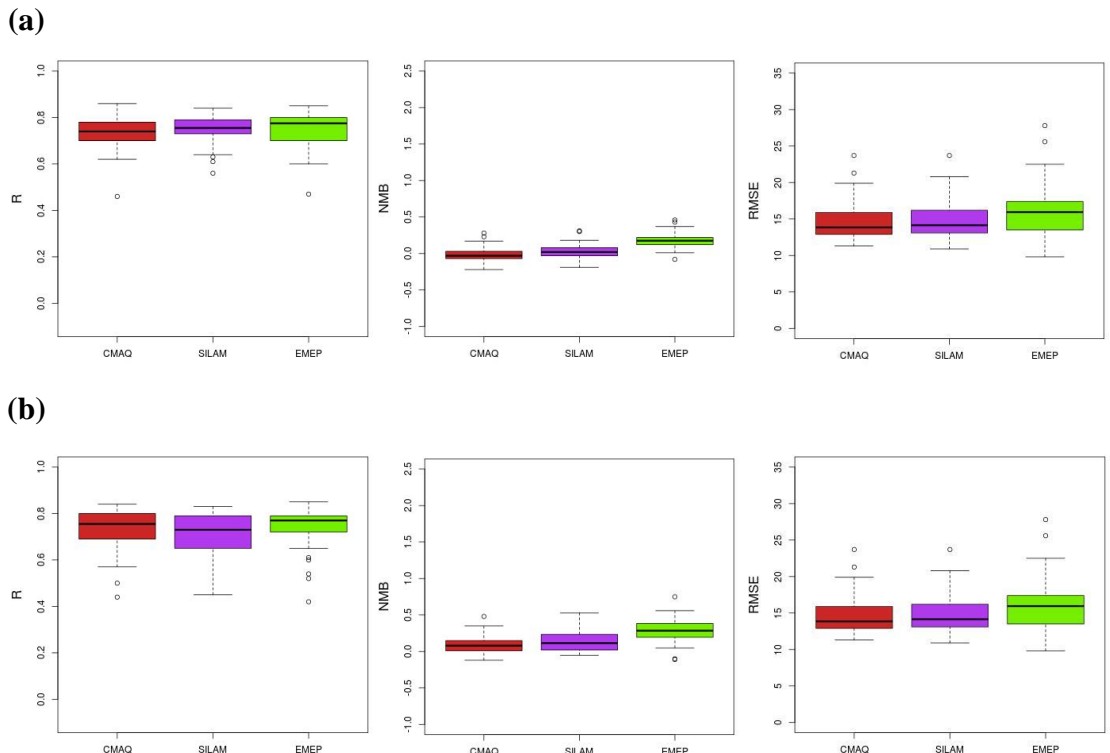

**Figure 4.** Comparison of statistical indicators for O$_3$ daily means (in the order R, NMB, and RMSE) between three CTMs at: (a) rural background stations and (b) urban background stations. Outlier shown as small circles.





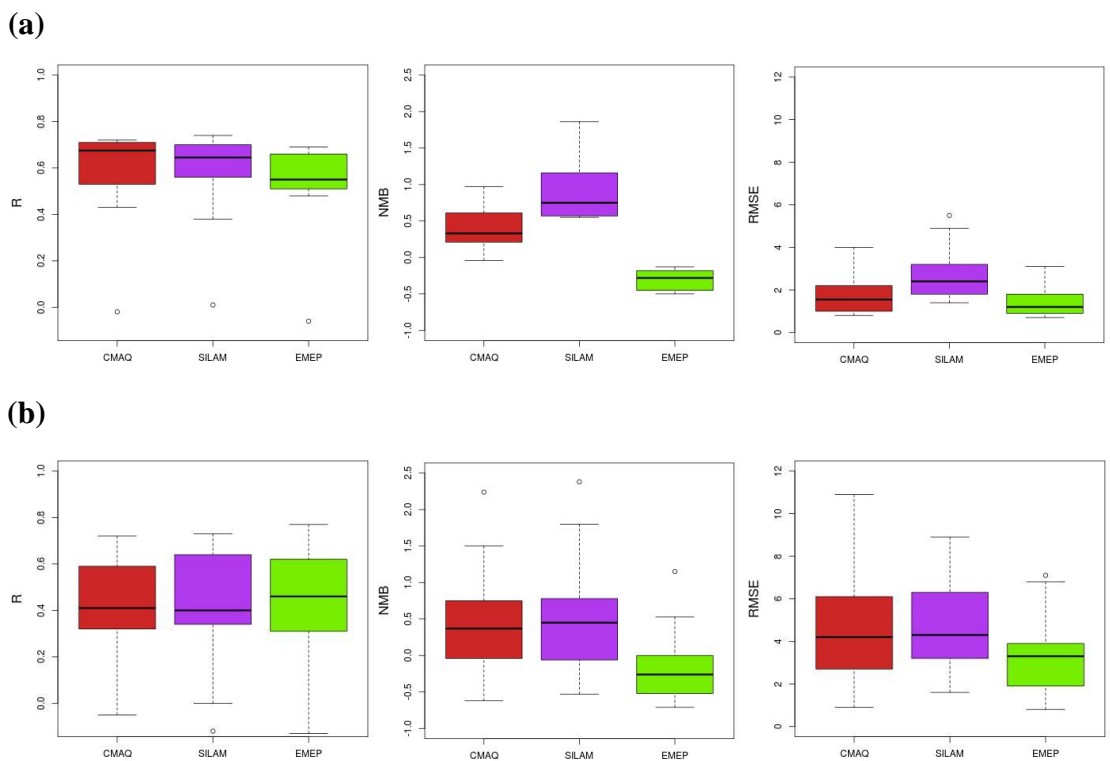

**Figure 5.** Comparison of statistical indicators for $SO_2$ daily means (in the order R, NMB, and RMSE) between three CTMs at: (a) rural background stations and (b) urban background stations. Outlier shown as small circles.





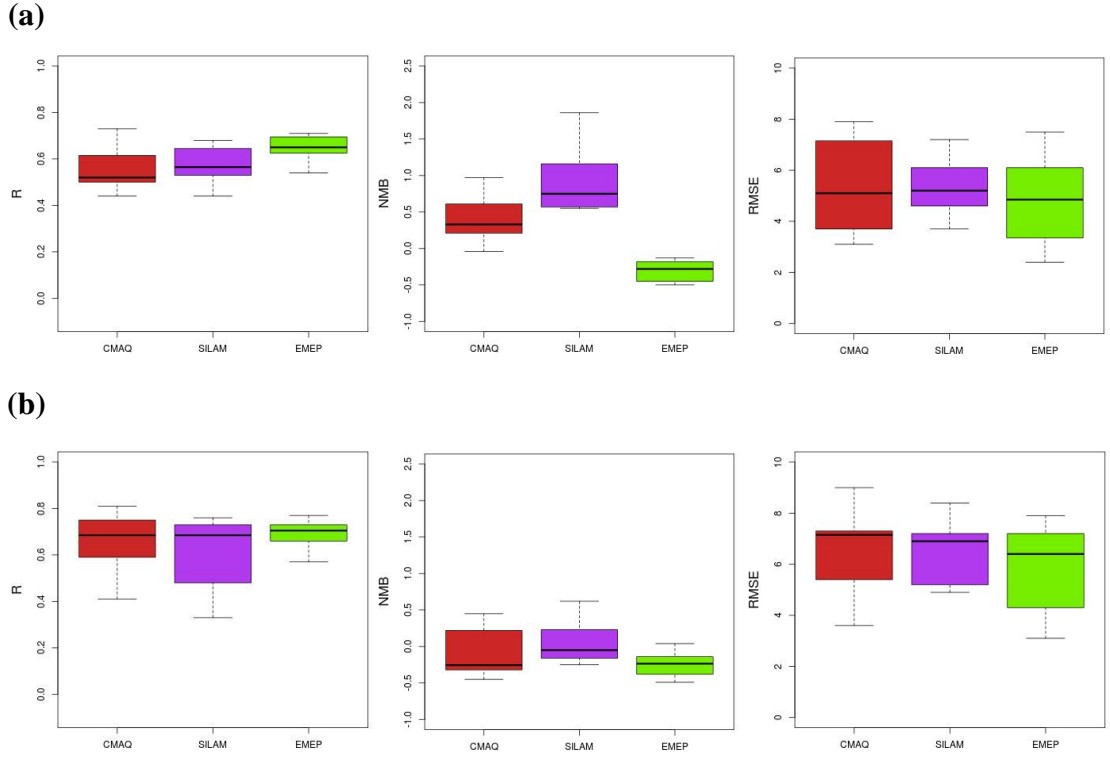

**Figure 6.** Comparison of statistical indicators for PM$_{2.5}$ daily means (in the order R, NMB, and RMSE) between three CTMs at: (a) rural background stations and (b) urban background stations. Outlier shown as small circles.





**Figure 7.** Comparison of the spatial distribution of annual mean concentrations ($\mu$g m$^{-3}$) from CMAQ (left column), SILAM (middle column) and EMEP (right column) in the Baltic Sea region for (a) O$_3$, (b) NO$_2$, (c) SO$_2$, and (d) PM$_{2.5}$. Empty areas correspond to concentrations between zero and the lowest value in the legend.





**Figure 8.** Comparison of the spatial distribution of annual mean ship-related concentrations (absolute ship contributions in µg m$^{-3}$) of the CMAQ (left column), SILAM (middle column) and EMEP (right column) models in the Baltic Sea region for (a) O$_3$, (b) NO$_2$, (c) SO$_2$, and (d) PM$_{2.5}$.



**Figure 9.** Comparison of PM$_{2.5}$ in summer and autumn from CMAQ (left column), SILAM (middle column) and EMEP (right column) in the Baltic Sea region for (a) JJA mean concentration (µg m$^{-3}$), (b) JJA mean ship contribution (µg m$^{-3}$), (c) SON mean concentration, and (d) SON mean ship contribution. Empty areas correspond to concentrations between zero and the lowest value in the legend.





**Figure 10.** Spatial distribution of the seasonal mean EC ship contribution (µg m$^{-3}$) from CMAQ (left column), SILAM (middle column) and EMEP (right column) in the Baltic Sea region for (a) spring, (b) summer, (c) autumn, and (d) winter. Empty areas correspond to concentrations between zero and the lowest value in the legend.





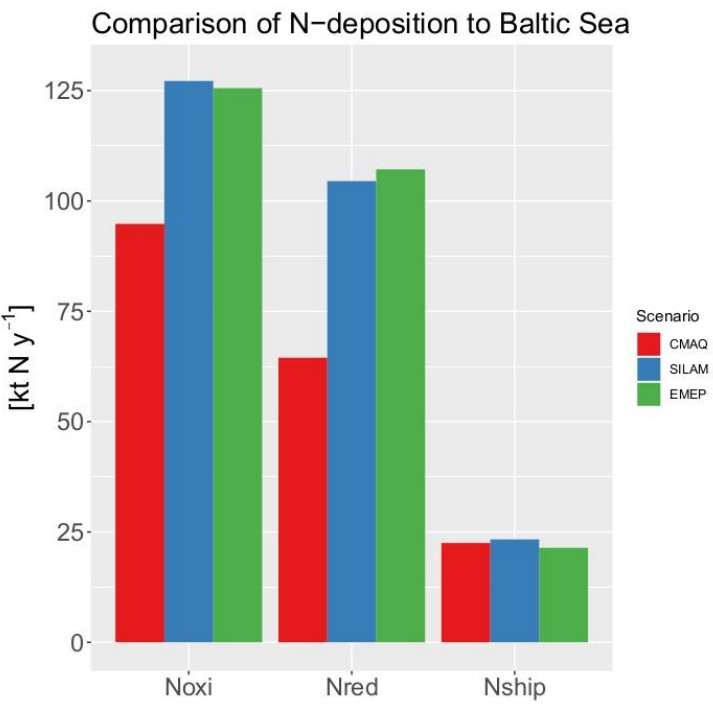

**Figure 11.** Comparison of the total annual deposition ($ktN\,y^{-1}$) of reduced and oxidised nitrogen and ship-related oxidised nitrogen ($N_{ship}$) to the Baltic Sea (seawater) from CMAQ (red), SILAM (dark blue) and EMEP/MSC-W (green).



**Table 1.** Description of the model setup of the three CTM systems.

| Model parameter | CMAQ | SILAM | EMEP/MSC-W |
|---|---|---|---|
| Horizontal grid resolution of the inner nest | $4 \times 4\,km^2$ | $4 \times 4\,km^2$ | $0.1° \times 0.1°$ |
| Nesting | D1: 64 km<br>D2: 16 km<br>D3: 4 km | D1: ~70 km<br>D2: ~18 km<br>D3: 4 km | D1: 0.1° |
| Meteorological driver | COSMO-CLM | WRF | IFS-ECMWF, Cy40r1 |
| Chemical boundary and initial conditions | FMI APTA global reanalysis | FMI APTA global reanalysis | Climatology for ozone |
| Land-based emissions | SMOKE-EU | SMOKE-EU | ECLIPSE |
| Ship emissions | STEAM $2 \times 2\,km^2$ | STEAM $2 \times 2\,km^2$ | STEAM $0.1° \times 0.1°$ |
| Ship emission time variability | hourly | hourly | monthly |

**Table 2.** Vertical profiles of ship emissions in the first seven vertical layers of the three CTM systems. The vertical distribution is given as average fractions of the total emission in each vertical model layer. Heights correspond to the heights of vertical layers in the models for a standard atmosphere.

| Layer nr. | CMAQ | | SILAM | | EMEP/MSC-W | |
|---|---|---|---|---|---|---|
| | Height | Fraction | Height | Fraction | Height | Fraction |
| 1 | 0–42 m | 0.24 | 0–20 m | 0.15 | 0–92 m | 1.00 |
| 2 | 42–84 m | 0.76 | 20–50 m | 0.13 | 92–184 m | 0 |
| 3 | 84–127 m | 0 | 50–100 m | 0.04 | 184–324 m | 0 |
| 4 | 127–180 m | 0 | 100–200 m | 0.08 | 324–525 m | 0 |
| 5 | 180–255 m | 0 | 200–300 m | 0.08 | 525–787 m | 0 |
| 6 | 255–340 m | 0 | 300–500 m | 0.15 | 787–1116 m | 0 |
| 7 | 340–430 m | 0 | 500–1000 m | 0.38 | 1116–1520 m | 0 |





**Table 3.** Statistical evaluation of daily mean PM$_{2.5}$ for the year 2012 with all three CTMs.

| Station code | N | $\mu_{Obs}$ | CMAQ | | | | | SILAM | | | | | EMEP/MSC-W | | | | |
|---|---|---|---|---|---|---|---|---|---|---|---|---|---|---|---|---|---|
| | | | $\mu_{Mod}$ | R | NMB | RMSE | FAC2 | $\mu_{Mod}$ | R | NMB | RMSE | FAC2 | $\mu_{Mod}$ | R | NMB | RMSE | FAC2 |
| DEHH008 | 363 | 12.6 | 9.8 | 0.73 | -0.23 | 7.3 | 0.73 | 11.1 | 0.68 | -0.12 | 7.2 | 0.86 | 10.8 | 0.71 | -0.14 | 7.5 | 0.87 |
| DEHH059 | 350 | 13.2 | 9.4 | 0.75 | -0.28 | 7.3 | 0.65 | 11.1 | 0.76 | -0.16 | 6.5 | 0.86 | 10.7 | 0.75 | -0.18 | 7.1 | 0.87 |
| DEMV019 | 364 | 12.9 | 7.1 | 0.79 | -0.45 | 8.1 | 0.42 | 9.7 | 0.69 | -0.25 | 7.9 | 0.77 | 8.0 | 0.73 | -0.38 | 7.9 | 0.59 |
| DEMV021 | 359 | 11.2 | 7.3 | 0.81 | -0.35 | 6.5 | 0.55 | 8.8 | 0.75 | -0.22 | 6.8 | 0.80 | 8.2 | 0.77 | -0.27 | 6.4 | 0.73 |
| DESH023 | 345 | 11.9 | 8.2 | 0.75 | -0.31 | 7.2 | 0.63 | 10.0 | 0.73 | -0.16 | 7.0 | 0.80 | 9.1 | 0.73 | -0.23 | 7.2 | 0.76 |
| FI00208 | 347 | 7.1 | 9.8 | 0.53 | 0.38 | 7.9 | 0.59 | 7.7 | 0.55 | 0.09 | 5.3 | 0.76 | 4.2 | 0.62 | -0.41 | 4.9 | 0.56 |
| FI00349 | 355 | 4.7 | 3.9 | 0.62 | -0.17 | 3.1 | 0.58 | 5.3 | 0.55 | 0.12 | 4.7 | 0.61 | 3.4 | 0.63 | -0.28 | 3.4 | 0.73 |
| FI00351 | 364 | 7.4 | 5.2 | 0.44 | -0.30 | 5.4 | 0.45 | 6.3 | 0.51 | -0.14 | 5.6 | 0.67 | 3.4 | 0.63 | -0.54 | 5.8 | 0.43 |
| FI00586 | 357 | 5.9 | 4.0 | 0.64 | -0.32 | 3.6 | 0.54 | 6.2 | 0.49 | 0.05 | 5.2 | 0.57 | 3.1 | 0.70 | -0.49 | 4.3 | 0.55 |
| FI00761 | 363 | 6.3 | 6.3 | 0.62 | 0.00 | 4.5 | 0.63 | 6.5 | 0.44 | 0.02 | 5.0 | 0.70 | 3.8 | 0.57 | -0.39 | 4.3 | 0.63 |
| FI00781 | 363 | 6.6 | 9.5 | 0.49 | 0.45 | 7.1 | 0.62 | 10.7 | 0.33 | 0.62 | 8.4 | 0.48 | 5.0 | 0.63 | -0.24 | 4.0 | 0.82 |
| LT00044 | 357 | 9.8 | 12.0 | 0.59 | 0.22 | 9.0 | 0.65 | 12.1 | 0.73 | 0.23 | 7.2 | 0.75 | 8.9 | 0.69 | -0.09 | 6.4 | 0.78 |
| SE0001A | 338 | 11.0 | 6.4 | 0.61 | -0.42 | 7.0 | 0.49 | 7.9 | 0.64 | -0.28 | 6.6 | 0.67 | 6.9 | 0.71 | -0.37 | 6.4 | 0.64 |
| SE0011R | 340 | 6.9 | 5.7 | 0.51 | -0.17 | 4.8 | 0.57 | 7.0 | 0.65 | 0.01 | 5.1 | 0.75 | 5.8 | 0.68 | -0.16 | 4.8 | 0.73 |
| SE0012R | 364 | 5.4 | 4.1 | 0.50 | -0.24 | 3.8 | 0.52 | 5.4 | 0.58 | -0.01 | 3.7 | 0.74 | 3.5 | 0.67 | -0.36 | 3.3 | 0.64 |
| SE0022A | 358 | 4.8 | 6.7 | 0.41 | 0.39 | 5.4 | 0.67 | 6.9 | 0.48 | 0.42 | 4.9 | 0.66 | 5.0 | 0.66 | 0.04 | 3.1 | 0.86 |
| SE0066A | 345 | 3.3 | 4.1 | 0.50 | 0.25 | 3.6 | 0.58 | 5.3 | 0.44 | 0.60 | 4.5 | 0.47 | 3.2 | 0.54 | -0.04 | 2.4 | 0.66 |

**Table 4.** Spatial averages of the annual mean concentrations of NO$_2$, O$_3$, SO$_2$, PM$_{2.5}$ and EC in µg m$^{-3}$ for the study domain (Baltic Sea region as in Fig. 1).

| CTM | NO$_2$ | O$_3$ | SO$_2$ | PM$_{2.5}$ | EC |
|---|---|---|---|---|---|
| CMAQ | 3.49 | 54.1 | 1.11 | 4.84 | 0.16 |
| SILAM | 4.42 | 57.6 | 1.61 | 5.95 | 0.15 |
| EMEP/MSC-W | 3.00 | 66.6 | 0.56 | 4.01 | 0.13 |



**Table 5.** Relative ship contribution to the spatial average of annual mean $NO_2$, $O_3$, $SO_2$, $PM_{2.5}$ and EC in percent for the study domain (Baltic Sea region as in Fig. 1).

| CTM | $NO_2$ | $O_3$ | $SO_2$ | $PM_{2.5}$ | EC |
|---|---|---|---|---|---|
| CMAQ | 28.3 | -0.4 | 14.5 | 6.5 | 7.3 |
| SILAM | 21.9 | 1.2 | 10.0 | 4.3 | 5.6 |
| EMEP/MSC-W | 21.8 | -0.1 | 19.1 | 5.7 | 5.3 |

**Table 6.** Spatial averages of seasonal mean concentrations of EC ($\mu g\,m^{-3}$) from the "base" simulation and ship contributions to EC levels ($\mu g\,m^{-3}$) for the study domain (Baltic Sea region as in Fig. 1). Mean values are given for spring (March to May; MAM), summer (JJA), autumn (Sept. to Nov.; SON) and winter (January, February and December 2012; DJF)

| Contribution | CTM | MAM | JJA | SON | DJF |
|---|---|---|---|---|---|
| | CMAQ | 0.134 | 0.081 | 0.154 | 0.277 |
| All emissions | SILAM | 0.136 | 0.109 | 0.183 | 0.166 |
| | EMEP/MSC-W | 0.111 | 0.072 | 0.150 | 0.191 |
| | CMAQ | 0.013 | 0.014 | 0.009 | 0.008 |
| Ship emissions | SILAM | 0.007 | 0.011 | 0.007 | 0.003 |
| | EMEP/MSC-W | 0.008 | 0.009 | 0.005 | 0.004 |