# Peer review of "Effects of ship emissions on air quality in the Baltic Sea region simulated with three different chemistry transport models"

_Atmospheric Chemistry and Physics, 2018_

## Referee Comment (RC1) · Anonymous Referee #1 · 29 Jan 2019

Dear Editor,

this is an interesting and useful paper comparing the performance of 3 CTMs for overall pollutant concentrations and also for shipping emissions. The first half of the MS deals with overall pollutant concentatrions, and the second with ship-sourced emissions. Several major aspects need to be addressed, in my opinion, prior to publication: 1) the MS should be shortened, as some sections are repetitive and feel like a report; 2) I would suggest to change the Figures and add correlation plots to show more of the data used for model validation, which are now not evident in the MS; 3) model validation of the shipping contributions, was it carried out? it looks like no validation

was performed, and this would need to be added, even if briefly; 4) if possible, add recommendations for users as to which model performs better under which scenarios.

Specific comments: - p1, l12-13: contradiction, is the performance of the models similar or does it differ for PM2.5 in summer?

- Introduction: can be shortened, specifically the paras dealing with CLs, literature review and SHEBA.

- p5, l23: are these total pollutant concentrations, or the ship-sourced fraction? This should be clarified throughout the text

- p6, l11-14: what about the non-linearity of O3? This approach (removing a source completely) has been seen to have higher uncertainty than if the source is only partially removed (e.g., decreasing its contribution by a given %), given that complete removal of the source doesn't take into account the non-linearity of certain species (e.g., O3). Please discuss how this may have affected the results.

- p8, l28: the use of monthly averaged gridded emissions is indeed a major difference between the models; wouldn't it have an impact also on the underestimation of titration, as described above for the spatial resolution (p4, l5-10)?

- The paper is very well referenced, in general.

- p11, l11: the model results were validated for total pollutant concentrations(e.g., against Airbase observation), and for ship-sourced pollutant concentrations (in this case, against what?)? Or only for total concentrations? PLease clarify.

- p11, Fig S1: it is not practical for the reader to start the Results section with Figs which are in Supporting Info. Maybe the authors could add the 4 time series at the bottom of Fig 3, for example?

- p11, l27: stations were grouped as rural and urban background, why? It would be more useful to see the individual points, instead of the averages, to have additional

Interactive
comment

detail.

- p13, l12: why were these 2 rural stations selected? Please clarify the criteria, here and for other pollutants.

- also in this section, instead of selecting 4 sites, what about plotting all of them in a correlation plot, for example the summer and winter mean per site? This would be helpful because with the current boxplot it is not so easy to see whether there is under or overestimation.

- subsections in section 3.1: their structure sounds a bit too much like a report, they are too similar (only changing the pollutant). Suggestion to redraft and shorten.

- section 3.1.3 (SO2); what is the reason for the poorer performance of models for SO2 at rural sites? There are larger differences between models, too. Please provide an explanation.

- p19, l4: recommendation to add a short concluding section on the comparison between models? This could include recommendations for users as to which model to select depending of the input data available or the purpose of the study.

- section 3.3: are the modelling contributions of shipping emissions validated in any way? A comparison with point locations could be carried out, based on literature review (even if the observational data correspond to different years, a qualitative comparison would still be necessary). Source apportionment studies should be used for this validation.

- p19, l12: would it be possible to provide an average for coastal areas? Or a range? This is usually where most population is exposed. This would be very useful for all the other pollutants as well.

- p21, l31: the ship-related EC concentrations are really low, were these data validated in any way? If not, please state clearly.

- section 3.6: the deposition section doesn't seem to fit in this MS, could it be included in the Karl et al. companion paper, instead? Otherwise, suggestion to remove it.

- p23, l25: the uncertainties of the models or or atmospheric transport and transformation of pollutants were not addressed in the MS; please add this in the new section on recommendations and conclusions, or remove this phrase.

—————————————————————

---

## Editor Comment (EC1) · Huan Liu (Editor) · 14 Mar 2019

This manuscript discussed air quality impacts using three models. The uncertainties of emissions, meteorology, chemical reactions and physical processes were discussed in detail. Overall, this is a good paper. Some technical questions:

1. The CMAQ version is too old to include advanced SOA mechanisms. Wildfire emissions were not included in emission inventory. All of above could be the reasons for lower estimation on summer SOA. These disadvantages should be fixed or at least discussed.

[Figure]

2. Authors should add the model validation for meteorology parameters.

3. Table 2, how do you get the "average fractions of the total emission in each vertical model layer"? This factor and its source need a very detailed description. Why the highest emission could reach 1000m in SILAM model? If this is true, the deposition process would be influenced a lot.

4. The references and equations for NMB, R, RMSE and FAC2 should be added.

5. The last sentence in section 3.3.2 is not accurate. It should be "NOx-limited regime in the model".
* * *

---

## Referee Comment (RC2) · Huan Liu (Referee) · 18 Mar 2019

This manuscript discussed air quality impacts using three models. The uncertainties of emissions, meteorology, chemical reactions and physical processes were discussed in detail. Overall, this is a good paper. Some technical questions:

1. The CMAQ version is too old to include advanced SOA mechanisms. Wildfire emissions were not included in emission inventory. All of above could be the reasons for low estimation on summer SOA. These disadvantages should be fixed or at least discussed.

[Figure]

2. Authors should add the model validation for meteorology parameters.

3. Table 2, how do you get the "average fractions of the total emission in each vertical model layer"? This factor and its source need a very detailed description. Why the highest emission could reach 1000m in SILAM model? If this is true, the deposition process would be influence a lot.

4. The references and equations for NMB, R, RMSE and FAC2 should be added.

5. The last sentence in section 3.3.2 is not accurate. It should be "NOx-limited regime in the model".
* * *

---

## Author Comment (AC3) · 15 Apr 2019

We refer to our Response to the (identical) Editor Comment under RC2.

———————————————

---

## Editor Comment (EC2) · Huan Liu (Editor) · 16 Apr 2019

Authors' responses could address my concern. I have no more questions.

———————————————

---

## Author Response (AR1)

**Changes to manuscript ms-nr acp-2018-1317**

**Effects of ship emissions on air quality in the Baltic Sea region simulated with three different chemistry transport models**

Matthias Karl (1,*), Jan Eiof Jonson (2), Andreas Uppstu (3), Armin Aulinger (1), Marje Prank (3,4), Mikhail Sofiev (3), Jukka-Pekka Jalkanen (3), Lasse Johansson (2), Markus Quante (1), Volker Matthias (1)

[1] Institute of Coastal Research, Helmholtz-Zentrum Geesthacht, 21502 Geesthacht, Germany.

[2] Norwegian Meteorological Institute, Oslo, Norway.

[4] Atmospheric Composition Research, Finnish Meteorological Institute, P.O. Box 503, FI-00101 Helsinki, Finland.

[3] now at: Cornell University, Ithaka, NY, U.S.A..

**Dear Dr Huan Liu,**

We highly appreciate the reviews of our manuscript ms-nr acp-2018-1317 that we received from you and one anonymous referee. We have replied to their comments in the Open Discussion. We have addressed all specific comments in the revised manuscript as will be described below. We carefully considered the concerns expressed by the anonymous referee and in the Editor Comment in our revision of the manuscript.

Below follows: (1) the point-by-point replies to the two reviewers, (2) a list of relevant changes in the manuscript, and (3) the revised manuscript with changes highlighted.

Our responses to reviewers have been written in blue font.

Figure, table, section, and page numbers in the replies below refer to the original manuscript. The revised manuscript with changes highlighted has been sent along with this response.

**Referee #1**

1. a) Several major aspects need to be addressed, in my opinion, prior to publication: 1) the MS should be shortened, as some sections are repetitive and feel like a report;

The manuscript has been shortened by significantly reducing the text of the Introduction, shortening the comparison with measurements, and removing the section on nitrogen deposition results. In particular, Sect. 3.1 ("Comparison to observations") was completely restructured to avoid the report-style text blocks.

1. b) 2) I would suggest to change the Figures and add correlation plots to show more of the data used for model validation, which are now not evident in the MS;

Figures showing time series plots of NO2 at coastal sites have been included in the new Appendix B. Time series plots of O3 at coastal sites are included in the SI, while all other time plots were removed. A new figure showing the spatial correlation (scatterplots with individual station points) as additional analysis of the model performance was included. We have kept the overview boxplots (Figures 3 - 6 in the original manuscript) because they give a compact overview of the three CTMs within one plot.

1. c) 3) model validation of the shipping contributions, was it carried out? it looks like no validation was performed, and this would need to be added, even if briefly;

This is not true. Although a direct comparison of the ship contribution was not carried out due to the methodological discrepancies with ship plume measurements (more explanations below), an evaluation method for the ship contribution had already been included in the original manuscript (Sect. 2.3.2 ""Significance of the ship contribution"). Following the comments of this Reviewer, we have revised the previous method for testing the significance of the ship contribution. We present the evaluation in the new Sect. 3.2 "Evaluation of ship-related concentration contributions" of the revised manuscript.

1. d) 4) if possible, add recommendations for users as to which model performs better under which scenarios.

We have added a new section 3.3.5 on recommendations for model users in the revised manuscript.

2. Specific comments: - p1, l12-13: contradiction, is the performance of the models similar or does it differ for PM2.5 in summer?

The sentence has been removed in the Abstract and also in the Conclusions (p. 24, line 1).

3. Introduction: can be shortened, specifically the paras dealing with CLs, literature review and SHEBA.

Introduction has been shortened in the revised manuscript.

4. p5, l23: are these total pollutant concentrations, or the ship-sourced fraction? This should be clarified throughout the text

This has been clarified here and at the other relevant places in the text. The title of Sect. 2.3.1 has been changed to "Evaluation method for the total air pollutant concentrations". The title of Sect. 3.1 has been changed to "Statistical evaluation of air pollutant concentrations". The significance of the ship contribution to total NO2 concentrations at monitoring stations has been evaluated in Sect. 3.1.5. This section has been renumbered as new Sect. 3.2 with title: "Evaluation of ship-related concentration contributions".

5. p6, l11-14: what about the non-linearity of O3? This approach (removing a source completely) has been seen to have higher uncertainty than if the source is only partially removed (e.g., decreasing its contribution by a given %), given that complete removal of the source doesn't take into account the non-linearity of certain species (e.g., O3). Please discuss how this may have affected the results

The procedure of deducing the ship contribution from one run including all emissions and one run without the emissions from shipping (zero-out method), which is used here, assumes linearity. The perturbation of the ship emissions, for example reduction by 20 %, as suggested by the Reviewer might be more careful with respect to the non-linearity of the involved photochemistry. However, our goal was to derive the full impact of shipping, while perturbing is mainly used to investigate the response to small changes (e.g. 20 %) of sectoral emissions. The assumption of linearity is reasonable. Moreover, the influence of shipping on ozone at coastal sites was found to be small. The following was added on p. 6 line 14:
"Previous calculations have shown that the assumption of linearity, by adding the contributions from different emission sources, is reasonable for ozone and other pollutants, and that the associated error is within a few percent (Jonson et al., 2018a; Karl et al., 2019)."

Jonson, J., Gauss, M., Schulz, M., and Nyíri, A.: Emissions from international shipping, in: Transboundary particulate matter, photo-oxidants, acidifying and eutrophying components, EMEP Status Report 1/2018, pp. 83-89, Norwegian Meteorological Institute, Oslo, Norway, 2018a.

Karl, M., Bieser, J., Geyer, B., Matthias, V., Jalkanen, J.-P., Johansson, L., and Fridell, E.: Impact of a nitrogen emission control area (NECA) on the future air quality and nitrogen deposition to seawater in the Baltic Sea region, Atmos. Chem. Phys., 19, 1721-1752, https://doi.org/10.5194/acp-19-1721-2019, 2019.

6. - p8, l28: the use of monthly averaged gridded emissions is indeed a major difference between the models; wouldn't it have an impact also on the underestimation of titration, as described above for the spatial resolution (p4, l5-10)?.

The use of monthly averaged ship emissions in the EMEP model, versus daily emissions, is explained in Sect. 2.2.5. Initial tests had been performed with the EMEP model using both daily

aggregated and monthly aggregated ship emissions of STEAM (from FMI). The differences of the model results were small, including for ozone. The statistical evaluation with ozone concentration measurements showed hardly any differences between the two setups. In particular, differences at coastal sites were of the order of +/- 0.01 ppb or less.

7. The paper is very well referenced, in general.

Thank you.

8. p11, l11: the model results were validated for total pollutant concentrations(e.g., against Airbase observation), and for ship-sourced pollutant concentrations (in this case, against what?)? Or only for total concentrations? PLease clarify.

We clarify that Sect. 2.3.1 ("Evaluation method for the total air pollutant concentrations") describes the method for evaluation of the total pollutant concentration and Sect. 2.3.2 ("Significance of the ship contribution") describes the method for evaluation of the ship contribution to the observed total concentration. The test of the significance of the ship influence at the monitoring stations can be regarded as an evaluation of the ship-related concentration, since it demonstrates how much the prediction of observed concentrations improves when shipping emissions are included in the simulation. The significance test has been repeated and the results of the evaluation are presented in the new Sect. 3.2 ("Evaluation of ship-related concentration contributions"). Including ship emissions improves the agreement between modelled and measured total NO2 daily mean concentrations at about 50 % of the stations.

9. p11, Fig S1: it is not practical for the reader to start the Results section with Figs which are in Supporting Info. Maybe the authors could add the 4 time series at the bottom of Fig 3, for example?

We assume that this point refers to page 13 (first reference to Fig. S1). Section 3.1 has been restructured in accordance with the next points of this Reviewer. The time series plots for the selected two rural and two urban sites were removed from the SI (i.e. Fig. S1 - S4).

The comparison has been merged with the analysis of the ship contribution at coastal stations. We therefore discuss the stations at shoreline and harbour cities in new Sect. 3.2 "Evaluation of ship-related concentration contributions" (Sect. 3.1.5 in the original manuscript). Time series plots of NO2 at coastal sites are now in the Appendix B. Time series plots of O3 at coastal sites are now in the new Fig. S1 of the SI.

10. p11, l27: stations were grouped as rural and urban background, why? It would be more useful to see the individual points, instead of the averages, to have additional detail.

The AirBase observation database has two main categories: rural background stations and urban background stations. Traffic and industrial sites were excluded since regional CTM systems applied in this study do not handle local scale dispersion (see p. 11, lines 29 - 32). Statistics for temporal correlations between modelled and observed total concentrations at individual stations are given in the SI (Tables S3 - S6). However, we now included the spatial

correlation (scatterplots with individual station points) as additional analysis of the model performance.

11. p13, l12: why were these 2 rural stations selected? Please clarify the criteria, here and for other pollutants.

The time series plots for the selected two rural and two urban sites were removed from the SI (i.e. Fig. S1 - S4) because the choice of the stations was arbitrary and the discussion of the time series plots did not provide additional information to the model evaluation. Instead, a spatial correlation analysis was added, see next point.

12. also in this section, instead of selecting 4 sites, what about plotting all of them in a correlation plot, for example the summer and winter mean per site? This would be helpful because with the current boxplot it is not so easy to see whether there is under or overestimation.

As recommended by this Reviewer, we have added scatterplots for the analysis of the spatial correlation of the annual mean total pollutant concentration together with the seasonal averages in the new Figure 7 and discuss this in the new subsection 3.1.2 "Spatial correlation". However, we prefer to keep the overview boxplots (Figures 3 - 6) because they give a compact overview and comparison of the statistics of the daily mean values from three CTMs within one plot.

13. subsections in section 3.1: their structure sounds a bit too much like a report, they are too similar (only changing the pollutant). Suggestion to redraft and shorten.

We followed the suggestion of this Reviewer to redraft and shorten Sect. 3.1. Table 3 has been moved to the Supplement. Information on statistical indicators can be seen in the boxplot figures, Fig. 3-6 (R, NMB and RMSE) and in the SI Tables (Tables S3-S6). Therefore, we have removed this information from the text. Sect. 3.1 has been restructured into the following subsections: 3.1.1 Rural versus urban sites and 3.1.2 Spatial correlation.

14. section 3.1.3 (SO2); what is the reason for the poorer performance of models for SO2 at rural sites? There are larger differences between models, too. Please provide an explanation.

The poorer performance of the models at the rural sites is related to uncertainties of local residential heating emissions. The following has been added on p. 15, line 29 of the original manuscript:

"The weaker performance of the models for SO2 at the rural sites is related to uncertainties of local residential heating emissions, as the timing of use and the sulfur content of burned fuels are difficult to predict."

15. p19, l4: recommendation to add a short concluding section on the comparison between models? This could include recommendations for users as to which model to select depending of the input data available or the purpose of the study.

Much of the discrepancies between the CTMs depend on the model configuration: ship emission, land-based emission, meteorology, and boundary conditions. We have added a new section 3.3.5 ("Recommendations from the comparison between the CTM systems") where we briefly evaluate the three models in terms of input data requirements, required level of user experience and model performance based on experience from this comparison. In addition, we give recommendations for which type and purpose of the study each model is suited best.

16. section 3.3: are the modelling contributions of shipping emissions validated in any way? A comparison with point locations could be carried out, based on literature review (even if the observational data correspond to different years, a qualitative comparison would still be necessary). Source apportionment studies should be used for this validation.

The evaluation of ship-related concentrations (of NO2) had been presented in Sect. 3.1.5 of the original manuscript. As mentioned in response to previous points of this Reviewer, we have revised the test of the significance of the ship contribution. We present the evaluation in the new Sect. 3.2 "Evaluation of ship-related concentration contributions". The evaluation of the ship contribution is only done for NO2 (daily means), since ship emissions are known to be a relevant contributor to ambient NO2 concentrations and NO2 is monitored at many stations in the coastal regions. In our view, it is not possible to compare the modelled ship-related concentration directly to measurements of the ship contribution at point locations. Usually, these point observations report the concentrations in the plume from a single ship (or a few ships), passing the site, in exceedance of the measured background value. In the models, emissions from all ships at sea within one hour and within a radius of up to 50 km upwind contribute to the ship signal at a point location. In particular for PM2.5 this can lead to large discrepancies between the two methods: during atmospheric transport of emitted pollutants, oxidation and condensation happens, leading to a high fraction of secondary aerosol to the PM2.5 signal, whereas point measurements mainly capture the contribution of primary particles to the PM2.5 signal. Performing source apportionment studies with the models is out of scope of the paper because the main goal is to quantify the effect of shipping emissions. Source apportionment would require tagging of all possible emission sectors that contribute to the total concentration at a point location. In addition, emissions from other sectors are associated with uncertainties, which might be even higher than that of shipping.

17. p19, l12: would it be possible to provide an average for coastal areas? Or a range? This is usually where most population is exposed. This would be very useful for all the other pollutants as well.

The spatial averaged ship contribution in the coastal land areas was added in Table 5, Table 6 and Table S10. The information on the average for coastal areas was also included in the manuscript text and the abstract.

18. p21, l31: the ship-related EC concentrations are really low, were these data validated in any way? If not, please state clearly.

The relatively low values for the modelled ship-related EC concentrations appear to be justified based on a comparison with data measured at a shoreline location in southern Sweden, which

report even lower contributions from ships. Despite the limitations mentioned in our reply to the previous point on evaluating modelled ship contributions, such a comparison gives some guidance regarding the plausibility of the modelled EC values. The following has been added to the manuscript (p.21, line 31):

"Measurements of the ship contribution to equivalent black carbon (eBC) concentrations at a shoreline location in southern Sweden (Falsterbo [55.3843 N, 12.8164 E] downwind of main shipping lanes, based on 113 individual plumes, reported a value of 0.0035 µg m-3 as average of the winter campaign in 2016 (Ausmeel et al., 2019). Wintertime average modelled ship-related EC at this location is factor 4 to 6 higher than the measured value (CMAQ: 0.0207 µg m-3; SILAM: 0.0144 µg m-3, EMEP/MSC-W: 0.0149 µg m-3). The discrepancy might arise from comparison with a different year than used in the model simulations. Another reason for the higher model values is that the CTMs consider all ships within a radius of 50 km upwind, whereas measurements considered individual ships passing by in a limited sea area."

Ausmeel, S., Eriksson, A., Ahlberg, E., and Kristensson, A: Methods for identifying aged ship plumes and estimating contribution to aerosol exposure downwind of shipping lanes, Atmos. Meas. Tech. Discuss., https://doi.org/10.5194/amt-2018-445, in review, 2019.

19. section 3.6: the deposition section doesn't seem to fit in this MS, could it be included in the Karl et al. companion paper, instead? Otherwise, suggestion to remove it.

Section 3.7 ("Comparison of oxidized nitrogen deposition") and Figure 11 have been removed.

20. p23, l25: the uncertainties of the models or atmospheric transport and transformation of pollutants were not addressed in the MS; please add this in the new section on recommendations and conclusions, or remove this phrase

The sentence has been removed.

**Editor Comment**

1. The CMAQ version is too old to include advanced SOA mechanisms. Wildfire emissions were not included in emission inventory. All of above could be the reasons for low estimation on summer SOA. These disadvantages should be fixed or at least discussed.

The aerosol scheme AERO5 was applied in the CMAQ model, considering the SOA formation pathways based on traditional two-product representation, including reaction of volatile organic compounds to give non-volatile products, oxidative ageing of primary organic aerosol, acid-catalysed enhancement of SOA mass, oligomerization reactions and in-cloud aqueous-phase oxidation. In CMAQ v5.2, the aerosol scheme AERO6 with multi-generational aging chemistry was introduced. This version had not been available at the time when the CMAQ simulations for this study were performed (2016). Primary organic aerosol (POA) previously treated as non-volatile and non-reactive, can evaporate, oxidize, and re-condense to form SOA, which is known as multi-generational aging of primary organic aerosol (Robinson et al., 2007). The multi-generational aging chemistry for the semi-volatile POA configuration introduced in CMAQ v5.2 is derived from the approach of Donahue et al. (2012) which takes into account the functionalization and fragmentation of organic vapours upon oxidation. However, atmospheric SOA processes are still not fully understood and models have difficulties with prediction of SOA (Jimenez et al., 2009). A discussion of the disadvantages of the used CMAQ version has been added in section 3.4:

"The SOA formation mechanism in the applied version of CMAQ (i.e. v5.0.1) is probably not adequate for reproducing the summertime aerosol. Primary organic aerosol (POA), SOA and organic vapours in the atmosphere should be considered a dynamic system that constantly evolves due to multi-generation oxidation (Robinson et al., 2007). We note that multi-generational aging chemistry for the semi-volatile POA was introduced in CMAQ v5.2, based on the approach of Donahue et al. (2012) which considers the functionalization and fragmentation of organic vapours upon oxidation. In addition, wildfire emissions have not been considered in the simulation with CMAQ. Wildfires emit large quantities of organic material and are associated with high biogenic VOC emissions due to high temperature, leading to increased SOA formation (Lee et al., 2008)."

Donahue, N. M., Kroll, J. H., Pandis, S. N., and Robinson, A. L.: A two-dimensional volatility basis set - Part 2: Diagnostics of organic aerosol evolution, Atmos. Chem. Phys., 12(2), 615–634, doi:10.5194/acp-12-615-2012, 2012.

Jimenez, J. L., Canagaratna, M. R., Donahue, N. M., Prevôt, A. S. H., Zhang, Q., Kroll, J. H., DeCarlo, P. F., Allan, J. D., Coe, H., Ng, N. L., Aiken, A. C., Docherty, K. S., Ulbrich, I. M., Grieshop, A. P., Robinson, A. L., Duplissy, J., Smith, J. D., Wilson, K. R., Lanz, V. A., Hueglin, C., Sun, Y. L., Tian, J., Laaksonen, A., Raatikainen, T., Rautiainen, J., Vaattovaara, P., Ehn, M., Kulmala, M., Tomlinson, J. M., Collins, D. R., Cubison, M. J., Dunlea, J., Huffman, J. A., Onasch, T. B., Alfarra, M. R., Williams, P. I., Bower, K., Kondo, Y., Schneider, J., Drewnick, F., Borrmann, S., Weimer, S.,

Demerjian, K., Salcedo, D., Cottrell, L., Griffin, R., Takami, A., Miyoshi, T., Hatakeyama, S., Shimono, A., Sun, J. Y., Zhang, Y. M., Dzepina, K., Kimmel, J. R., Sueper, D., Jayne, J. T., Herndon, S. C., Trimborn, A. M., Williams, L. R., Wood, E. C., Middlebrook, A. M., Kolb, C. E., Baltensperger, U., and Worsnop, D. R.: Evolution of organic aerosols in the atmosphere, Science 326, 1525–1529, doi: 10.1126/science.1180353, 2009.

Lee, S., Kim, H. K., Yan, B., Cobb, C. E., Hennigan, C., Nichols, S., Chamber, M., Edgerton, E. S., Jansen, J. J., Hu, Y., Zheng, M., Weber, R. J., and Russell, A. G.: Diagnosis of aged prescribed burning plumes impacting an urban area, Environ. Sci. Technol., 42(5), 1438–1444 doi:10.1021/es7023059, 2008.

Robinson, A. L., Donahue, N. M., Shrivastava, M. K., Weitkamp, E. A., Sage, A. M., Grieshop, A. P., Lane, T. E. and Pierce, J. R., and Pandis, S. N.: Rethinking Organic Aerosols: semivolatile emissions and photochemical aging, Science 315, 1259–1262, 2007.

2. Authors should add the model validation for meteorology parameters.

Reviewer #1 has already asked to shorten the manuscript. Therefore, we refrain from adding a detailed evaluation of the meteorological fields used in each of the three CTMs. Regarding the evaluation of WRF used in the SILAM simulation we refer to the study by Kryza et al. (2017) where a WRF setup with similar configuration and spatial resolution has been evaluated with station measurements in Poland. For the EMEP model we refer to the evaluation of the ECMWF weather forecast from cycle Cy40r1 as summarized in Haiden et al. (2014). The meteorological fields from the COSMO-CLM model that were used in the CMAQ simulation have been evaluated with respect to frequency and amount of precipitation in Karl et al. (2019). We include an evaluation of the 2 m air temperature (T2) and the wind speed at 10 m (WS10) of the 0.025 degree COSMO-CLM data in the southern part of the Baltic Sea (BS) domain. Temperature was compared against gridded observational data from E-OBS v.16 (Cornes et al., 2018) of the European Climate Assessment & Dataset (ECA&D). Wind speed was compared against observational data from MiKlip DecReg of the German Weather Service (DWD). Monthly T2 in Denmark and southern Sweden (Fig. C1) was underestimated in winter (bias smaller than -1.4 K) and overestimated in summer. The warm bias in summer was higher in Sweden (+1.4 K) than in Denmark (+0.4 K). In contrast to southern Sweden, the winter T2 was overestimated in the more northern parts of Sweden (Fig. C2). The spatial correlation of T2 in the southern BS domain based on 3-daily averages was remarkably good (R = 0.94). Monthly WS10 was slightly overestimated in most parts of the southern BS domain (Fig. C3). The largest errors of wind speed occurred in Denmark and northern Poland during May and June.

The summary of the evaluation of meteorological variables has been added to section 2.2.2 ("Meteorology") in the revised manuscript.

Cornes, R., van der Schrier, G., van den Besselaar, E. J. M., and Jones, P. D.: An ensemble version of the E-OBS temperature and precipitation datasets, J. Geophys. Res. Atmos., 123, doi:10.1029/2017JD028200, 2018.

Haiden, T., Janousek, M., Bauer, P., Bidlot, J., Ferranti, L., Hewson, T., Prates, F., Richardson, D. S., and Vitart, F.: Evaluation of ECMWF forecasts, including 2013-2014 upgrades, ECMWF Technical Memorandum No. 742, December 2014, European Centre for Medium-Range Weather Forecasts, Reading, UK, 2014.

Karl, M., Bieser, J., Geyer, B., Matthias, V., Jalkanen, J.-P., Johansson, L., and Fridell, E.: Impact of a nitrogen emission control area (NECA) on the future air quality and nitrogen deposition to seawater in the Baltic Sea region, Atmos. Chem. Phys., 19, 1721-1752, https://doi.org/10.5194/acp-19-1721-2019, 2019.

Kryza, M., Walaszek, K., Ojrzynska, H., Szymanowski, M., Werner, M., and Dore, A. J.: High-resolution dynamical downscaling of ERA-Interim using the WRF regional climate model for the area of Poland. Part 1: Model configuration and statistical evaluation for the 1981-2010 period, Pure Appl. Geophys., 174(2), 511-526, doi:10.1007/s00024-016-1272-5, 2017.

a)

[Figure]

b)

Figure C1: Evaluation of COSMO-CLM data: temporal bias a) of the monthly mean 2 m air temperature in Denmark (left) and southern Sweden (right); and b) of the monthly mean 10 m wind speed in Denmark (left) and southern Sweden (right).

[Figure]

Figure C2: Evaluation of COSMO-CLM data: spatial distribution of the temporal bias of the monthly mean 2 m air temperature in the southern part of the Baltic Sea region.

[Figure]

Figure C3: Evaluation of COSMO-CLM data: spatial distribution of the temporal bias of the monthly mean 10 m wind speed in the southern part of the Baltic Sea region. Missing observation data for January 2012.

3. Table 2, how do you get the "average fractions of the total emission in each vertical model layer"? This factor and its source need a very detailed description. Why the highest emission could reach 1000m in SILAM model? If this is true, the deposition process would be influence a lot.

The average fractions of total emissions in each vertical model layer are derived from the two STEAM datasets, one containing ship emissions below 36 m and one containing ship emissions above 36 m (p. 11, line 3). The dataset with ship emissions above 36 m was intended to represent the emissions of ships with a stack exit above 36 m above sea level. Ship emissions from the two datasets were distributed into the different vertical layers of the CTMs. This had been done differently for the three models because of the different model layer heights. Unfortunately, further analysis of the SILAM results revealed, that the STEAM data for ship emissions above 36 m had been injected in the model's vertical layers between 36 m up to 1000 m height. We agree that this procedure was erroneous. Consequently, we have repeated the SILAM "base" run with ship emissions vertically distributed in the same kind as in the CMAQ model. New results from the SILAM run show higher ship contribution of NO2, SO2, and more ozone titration. Further, the statements about the effect of the vertical ship emission profile on ship-related EC concentrations of SILAM in Sect. 3.5 ("Comparison of elemental carbon related to ship emissions") have been removed. Table 2 has been removed because it is now unnecessary.

4. The references and equations for NMB, R, RMSE and FAC2 should be added.

Definitions of these statistical indicators are now given in an Appendix A.

5. The last sentence in section 3.3.2 is not accurate. It should be "NOx-limited regime in the model".

The results from the new SILAM "base" run show a similar tendency for annual mean O3 concentration changes due to shipping as the other two models. The referred sentence has therefore been removed.

**List of relevant changes in the ms**

**Relevant text changes:**

**1.     Introduction**

As requested by Referee #1, the Introduction has been shortened.

**2.     Section 2 "Description of the CTM systems and setup of the model comparison"**

Following the Comment from the Editor, we have added a brief evaluation of the meteorological fields used in each of the three CTMs in Sect. 2.2.2 ("Meteorology") (p.9, line 18):

"The SILAM model is run with meteorological input from a simulation with the Weather Research and Forecast (WRF) model v3.7.1 using original resolutions of 4.0 km, 16.0 km, and 64.0 km, for inner, central and outer domains, respectively. WRF was driven with large scale meteorological forcing data taken from the ERA-Interim reanalysis of ECMWF (Dee et al. 2011). The high-resolution inner domain extended up to 2000 m height and was therefore less influenced by upper tropospheric dynamics of WRF. Kryza et al. (2017) using WRF in a similar configuration evaluated the WRF meteorological fields against station observations in Poland. The 2 m air temperature (T2) was underestimated in winter (bias smaller than -0.6 K) while temperature in the warm season was overestimated (bias up to +1.0 K). The largest errors for the 10 m wind speed (WS10) occurred in late summer and autumn and the largest errors for wind direction (WDIR) in spring and summer. The error of wind direction was very small in winter. Spatial distribution of meteorological variables obtained from WRF were in close agreement with the station measurements, but the model performance was found to be worse for the seashore and mountain areas than for other inland areas (Kryza et al., 2017).

High-resolution meteorological fields for CMAQ were obtained from the COSMO-CLM (Rockel et al., 2008) model version 5.0. The meteorological fields were converted to the extension, resolution and projection of the CMAQ nested grids, using an in-house modified version of MCIP. More details on the meteorological forcing data and the evaluation of precipitation can be found in Karl et al. (2019). Here we include an evaluation of T2 and WS10 in the southern part of the Baltic Sea region. Temperature was compared against gridded observational dataset E-OBS v.16 (Cornes et al., 2018). Wind speed was compared against observational data from MiKlip DecReg of the German Weather Service (DWD). Monthly mean T2 in Denmark and southern Sweden was underestimated in winter (bias smaller than -1.4 K) and overestimated in summer. The warm bias in summer was higher in Sweden (+1.4 K) than in Denmark (+0.4 K). The spatial correlation of T2 in the southern Baltic Sea region based on 3-daily averages was remarkably good. Monthly mean WS10 was slightly overestimated in most parts of the region. The largest errors of wind speed occurred in Denmark and northern Poland during May and June.

EMEP/MSC-W was driven by meteorological data from the Integrated Forecasting System (IFS) of the ECMWF, version IFS38r2, with t1279 resolution (about 0.16 degrees resolution) interpolated to 0.1 × 0.1 degrees. The ECMWF forecasting system of weather parameters is regularly validated by

comparing against European synoptic observation data available on the Global Telecommunication System (GTS). The evaluation of the weather forecast from cycle Cy40r1 is summarized as follows (Haiden et al., 2014). The frequency of light precipitation is overestimated, with a bias of 1.2–1.4 mm d$^{-1}$ (for precipitation amounts > 1mm d$^{-1}$). T2 has a negative night-time temperature bias over Europe in winter and early spring. For total cloudiness, bias and standard deviation are small in 2012. For WS10, the standard deviation is low and the night-time bias is very small."

Sect. 2.3 ("Statistical analysis") has been modified to make it clearer that Sect. 2.3.1 ("Evaluation method for the total air pollutant concentrations") describes the method for evaluation of the total pollutant concentration and Sect. 2.3.2 ("Significance of the ship contribution") describes the method for evaluation of the ship contribution to the observed total concentration. The test of the significance of the ship influence at the monitoring stations can be regarded as an evaluation of the ship-related concentration, since it demonstrates how much the prediction of observed concentrations improves when shipping emissions are included in the simulation. As requested by the Editor, the statistical indicators NMB, R, RMSE and FAC2 are now defined in a new Appendix A.

**3. Comparison to observations**

The title of Sect. 3.1 has been changed to "Statistical evaluation of air pollutant concentrations". The significance of the ship contribution to total NO2 concentrations at monitoring stations has been evaluated in Sect. 3.1.5 in the original manuscript. This section has been renumbered as new Sect. 3.2 with title: "Evaluation of ship-related concentration contributions".

We followed the suggestion of Referee #1 to redraft and shorten Sect. 3.1. Sect. 3.1 has been restructured into the following subsections: 3.1.1 "Rural versus urban sites" and 3.1.2 "Spatial correlation".

Sect. 3.1.1 presents the statistical evaluation of the temporal variation of total air pollutant concentrations. The performance of the models to simulate air pollutant concentrations is compared and discussed separately for the group of rural stations and for the group of urban stations in order to highlight differences in the predictive capability of the models for rural versus urban sites.

The discussion of time series plots for the selected two rural and two urban sites were removed from the manuscript and the corresponding figures in the SI have been deleted (i.e. Fig. S1 - S4) because the choice of the stations was arbitrary and the discussion of the time series plots did not provide additional information to the model evaluation. Information on statistical indicators can be seen in the boxplot figures, Fig. 3-6 (R, NMB and RMSE) and in the SI Tables (Tables S3-S6). Therefore, we have removed this information from the text. We have kept the overview boxplots (Figures 3 - 6 in the original manuscript) because they give a compact overview of the three CTMs within one plot.

As recommended by Referee #1, we have added scatterplots for the analysis of the spatial correlation of the annual mean total pollutant concentration together with the seasonal averages in the new Figure 7 and discuss this in the new subsection 3.1.2 "Spatial correlation".

In the new Sect. 3.2 "Evaluation of ship-related concentration contributions", we discuss the stations at shoreline and harbour cities. Time series plots of $NO_2$ at coastal sites are now in the new Appendix B. Time series plots of $O_3$ at coastal sites are now in the new Fig. S1 of the SI.

Due to the new Sect. 3.2, the subsequent sections of the Results section have been renumbered.

**4.    Section 3.3 "Comparison of the spatial distribution of air quality indicators"**

Following a comment from Referee #1, we have added the spatial averaged ship contribution in the coastal land areas in Table 3 and 4 (Table 5 and 6 of the original manuscript) and Table S10, in addition to that for the entire Baltic Sea region (which included also the area of the sea). The information on the average for coastal areas was also included in the manuscript text and the abstract.

We have added a new section 3.3.5 ("Recommendations from the comparison between the CTM systems") where we briefly evaluate the three models in terms of input data requirements, required level of user experience and model performance based on experience from this comparison. In addition, we give recommendations for which type and purpose of the study each model is suited best.

**5.    Section 3.4 "Comparison of the ship contribution in the three CTMs"**

Following the comment of the Editor on the vertical ship emission distribution in the SILAM model, we have repeated the SILAM "base" run with ship emissions vertically distributed in the same kind as in the CMAQ model. The ship contribution calculated by the SILAM model to the ambient air pollutant has changed. Consequently, we have revised the text of Sect. 3.4 "Comparison of the ship contribution in the three CTMs" and the information in Table 3 (Table 5 of the original manuscript).

**6.    Section 3.6 "Comparison of elemental carbon related to ship emissions"**

Following the Editor Comment, a discussion of the disadvantages of the used CMAQ version with respect to SOA formation has been added in Sect. 3.5 "Comparison of $PM_{2.5}$ in summer and autumn" (p. 20, line 25):

"The SOA formation mechanism in the applied version of CMAQ (i.e. v5.0.1) is probably not adequate for reproducing the summertime aerosol. Primary organic aerosol (POA), SOA and organic vapours in the atmosphere should be considered a dynamic system that constantly evolves due to multi-generation oxidation (Robinson et al., 2007). We note that multi-generational aging chemistry for the semi-volatile POA was introduced in CMAQ v5.2, based on the approach of Donahue et al. (2012) which considers the functionalization and fragmentation of organic vapours upon oxidation. In addition, wildfire emissions have not been considered in the simulation with CMAQ. Wildfires emit large quantities of organic

material and are associated with high biogenic VOC emissions due to high temperature, leading to increased SOA formation (Lee et al., 2008).”

**7.      Section 3.6 “Comparison of elemental carbon related to ship emissions”**

The relatively low values for the modelled ship-related EC concentrations appear to be justified based on a comparison with data measured at a shoreline location in southern Sweden, which report even lower contributions from ships. The following has been added to the manuscript (p.21, line 31):

“Measurements of the ship contribution to equivalent black carbon (eBC) concentrations at a shoreline location in southern Sweden (Falsterbo [55.3843 N, 12.8164 E] downwind of main shipping lanes, based on 113 individual plumes, reported a value of 0.0035 µg m$^{-3}$ as average of the winter campaign in 2016 (Ausmeel et al., 2019). Wintertime average modelled ship-related EC at this location is factor 4 to 6 higher than the measured value (CMAQ: 0.0207 µg m$^{-3}$; SILAM: 0.0144 µg m$^{-3}$, EMEP/MSC-W: 0.0149 µg m$^{-3}$). The discrepancy might arise from comparison with a different year than used in the model simulations. Another reason for the higher model values is that the CTMs consider all ships within a radius of 50 km upwind, whereas measurements considered individual ships passing by in a limited sea area.”

Following the comment of the Editor on the vertical ship emission distribution in the SILAM model, we have repeated the SILAM “base” run with ship emissions vertically distributed in the same kind as in the CMAQ model. The discussion of the impact of vertical emission profile on ship-related EC concentration has therefore been removed.

**7.      Deposition of nitrogen**

Sect. 3.6 “Comparison of oxidised nitrogen deposition” of the original manuscript has been removed because of the concern expressed by Referee #1.

**8.      Summary and Conclusions**

Following the concern expressed by Referee #1, the sentence on p.23, line 25 (original manuscript) about the uncertainties of the models or atmospheric transport and transformation of pollutants has been deleted.

The conclusion on the evaluation of ship-related contribution to ambient levels of NO2 and O3 has been revised. The statements about nitrogen deposition (p.25, line 3-6) and vertical ship emission profile (p.25, line 11-14) have been removed.

**Tables:**

Table 2.

The table has been deleted.

Table 3.

The table has been moved to the Supplementary Information and is now Table S6.

Table 4.

The table has been renumbered as Table 2.

Table 5.

The table has been renumbered as Table 3.

Table 6.

The table has been renumbered as Table 4.

**Figures:**

Figure 7.

As recommended by Referee #1, we have added scatterplots for the analysis of the spatial correlation of the annual mean total pollutant concentration together with the seasonal averages in the new Figure 7.

Figure 7 of the original manuscript has been renumbered as Figure 8.

Figure 8.

The figure has been renumbered as Figure 9.

Figure 9.

The figure has been renumbered as Figure 10.

Figure 10.

The figure has been renumbered as Figure 11.

Figure 11.

The figure has been deleted.

Figure B1.

[revised manuscript text omitted]